# Targeting bacterial nickel transport with aspergillomarasmine A suppresses virulence-associated Ni-dependent enzymes

David Sychantha[1,2,3,5], Xuefei Chen[1,2,3], Kalinka Koteva [1,2,3], Gerd Prehna [4] & Gerard D. Wright [1,2,3] ✉

Microbial $Ni^{2+}$ homeostasis underpins the virulence of several clinical pathogens. $Ni^{2+}$ is an essential cofactor in urease and [NiFe]-hydrogenases involved in colonization and persistence. Many microbes produce metallophores to sequester metals necessary for their metabolism and starve competing neighboring organisms. The fungal metallophore aspergillomarasmine A (AMA) shows narrow specificity for $Zn^{2+}$, $Ni^{2+}$, and $Co^{2+}$. Here, we show that this specificity allows AMA to block the uptake of $Ni^{2+}$ and attenuate bacterial Ni-dependent enzymes, offering a potential strategy for reducing virulence. Bacterial exposure to AMA perturbs $H_2$ metabolism, ureolysis, struvite crystallization, and biofilm formation and shows efficacy in a *Galleria mellonella* animal infection model. The inhibition of Ni-dependent enzymes was aided by $Zn^{2+}$, which complexes with AMA and competes with the native nickelophore for the uptake of $Ni^{2+}$. Biochemical analyses demonstrated high-affinity binding of AMA-metal complexes to NikA, the periplasmic substrate-binding protein of the $Ni^{2+}$ uptake system. Structural examination of NikA in complex with Ni-AMA revealed that the coordination geometry of Ni-AMA mimics the native ligand, Ni-(L-His)$_2$, providing a structural basis for binding AMA-metal complexes. Structure-activity relationship studies of AMA identified regions of the molecule that improve NikA affinity and offer potential routes for further developing this compound as an anti-virulence agent.

$Ni^{2+}$ is a trace metal ion cofactor essential in the physiology of several microorganisms[1]. Only ten Ni-dependent enzymes are currently known in nature, of which urease and [NiFe] hydrogenase are the best studied because of their established correlation with microbial virulence[2–5]. The significance of these Ni-dependent enzymes is underscored by animal infection models linking the role of Ni-dependent enzymes to the colonization and survival of the bacteria *Staphylococcus aureus*[6,7], *Helicobacter pylori*[8,9], *Klebsiella pneumoniae*[10], *Proteus mirabilis*[11], and *Salmonella enterica* serovar Typhimurium[12,13],

and the fungi *Cryptococcus neoformans*[14,15] and *Aspergillus fumigatus*[16]. In contrast, no Ni-dependent enzymes have been identified in mammals. However, human calprotectin chelates $Ni^{2+}$ with high affinity as part of the innate immune response[17]. As a result, calprotectin prevents the uptake of $Ni^{2+}$ into bacteria, thereby attenuating urease[17]. This intrinsic mechanism of $Ni^{2+}$ limitation at the host-pathogen interface highlights the biological importance of this metal as an accessory to the pathophysiology of many microorganisms.

[1]David Braley Centre for Antibiotic Discovery, McMaster University, Hamilton, ON, Canada. [2]M.G. DeGroote Institute for Infectious Disease Research, McMaster University, Hamilton, ON, Canada. [3]Department of Biochemistry and Biomedical Sciences, McMaster University, Hamilton, ON, Canada. [4]Department of Microbiology, University of Manitoba, Winnipeg, MB, Canada. [5]Present address: Department of Chemistry, University of Waterloo, Waterloo, ON, Canada. ✉e-mail: wrightge@mcmaster.ca

**Fig. 1 | Nickel trafficking in *Klebsiella pneumoniae*.** 1) $Ni^{2+}$ is sequestered by L-histidine to form a 2:1 complex (Ni-(L-His)$_2$), passing through an unknown porin into the periplasm. 2) Ni-(L-His)$_2$ is bound by NikA once in the periplasm and transported into the cytoplasm via the NikBCDE ABC-type transporter. 3) In *K. pneumoniae*, cytoplasmic $Ni^{2+}$ is partitioned between two trafficking systems to mature Ni-dependent enzymes. UreDEFG and HypAB assemble the active site metallocenters of urease and the [NiFe]-hydrogenase (Hyd-3) of the formate hydrogenlyase complex, respectively. The colors represent different protein subunits and are not assigned to different functional classes.

Among urease-producing bacteria, *Klebsiella spp.*, *Proteus spp.*, *Staphylococcus spp.*, *Pseudomonas spp.*, and *Providencia spp.* are frequently associated with hospital-acquired urinary tract infections (UTIs)[18]. These pathogens are the etiological agents of struvite stones (magnesium ammonium phosphate; $NH_4MgPO_4 \cdot 6H_2O$), which account for ~15% of all urinary stones. Struvite accumulates on urinary catheters and the urothelial surface in the early stages of colonization, facilitating biofilm formation and persistence[19]. The mechanism by which these stones form involves urea hydrolysis to ammonia ($NH_3$) and carbon dioxide ($CO_2$) catalyzed by the enzyme urease. These hydrolysis products raise the pH of urine (pH >7.5), promoting the crystallization of struvite from naturally present magnesium and phosphate ions. Such alkalinization of biological fluids may also cause tissue damage and enable bacterial survival at low pH, exacerbating UTI symptoms and bacterial burden[18]. Furthermore, $NH_3$ liberated by urease has been shown to serve as a nitrogen source for *Klebsiella aerogenes*[20], implicating urease in host nutritional adaptation. Similarly, four distinct classes of [NiFe]-hydrogenases either oxidize hydrogen ($H_2$) for proton uptake or reduce protons to oxidize formate and evolve $H_2$, playing multifaceted roles in virulence and physiology[21]. More specifically, *H. pylori* and *S.* Typhimurium utilize $H_2$ for energy, promoting virulence, whereas *K. pneumoniae* evolves $H_2$, which can protect it from oxidative stress[12,22,23]. Given these examples, the enzymatic function of Ni-dependent enzymes plays a significant role in UTIs and other infectious diseases and is underpinned by the acquisition of an essential $Ni^{2+}$ cofactor.

The enzymatic activities of urease and [NiFe]-hydrogenases depend on the availability of $Ni^{2+}$ in the cytoplasm. The ABC-type transporter (NikABCDE) imports the metal from the extracellular environment through a multi-step process in many bacteria, including *Enterobacteriaceae* and *S. aureus*[6,24,25]. While a bacterial nickelophore dedicated to the NikABCDE system is currently unknown, exogenous environmental L-histidine is believed to be a widespread ligand. L-Histidine forms Ni-(L-His)$_2$ complexes, which exhibit a strong affinity for NikA for efficient transport into the cell through the NikBCDE ABC-type transporter (Fig. 1)[26,27].

Previously, we identified that the fungal-derived polycarboxylic acid natural product aspergillomarasmine A (AMA) is a non-antimicrobial metallophore with a preference for $Zn^{2+}$, $Co^{2+}$, and $Ni^{2+}$ ions[28–30]. AMA comprises L-Asp and two units of 2,3-diaminopropionic acid (APA) (Fig. 2A)[31]. Our previous work also showed that AMA sequesters $Zn^{2+}$ and inhibits $Zn^{2+}$-dependent metallo-β-lactamases (MBLs) by preventing their activation and promoting spontaneous $Zn^{2+}$ dissociation and instability[28,32]. This activity raised the possibility that AMA could similarly sequester extracellular $Ni^{2+}$ to prevent its uptake and block the maturation of urease and [NiFe]-hydrogenases. Since antibiotic-resistant UTIs are becoming increasingly common in healthcare settings, inhibiting $Ni^{2+}$ uptake could reduce complications and the societal burden associated with such infections[33].

In this work, we explore the potential of AMA to prevent $Ni^{2+}$ uptake and attenuate urease and hydrogenase activity in pathogenic bacteria. We show that both AMA and its metal complexes inhibited urease in *S. aureus* and *K. pneumoniae*, suppressing the crystallization of struvite in urine cultures. In addition, Zn-AMA is not toxic to mammalian cells and has in vivo efficacy in *Galleria mellonella* larvae. AMA-metal complexes target the periplasmic $Ni^{2+}$ uptake protein NikA through high-affinity interactions and compete with Ni-(L-His)$_2$ for uptake. Co-crystallographic analysis of NikA and Ni-AMA revealed that Ni-AMA resembles Ni-(L-His)$_2$, providing a structural basis for uptake. Finally, we explored the structure-activity relationships of Zn-AMA,

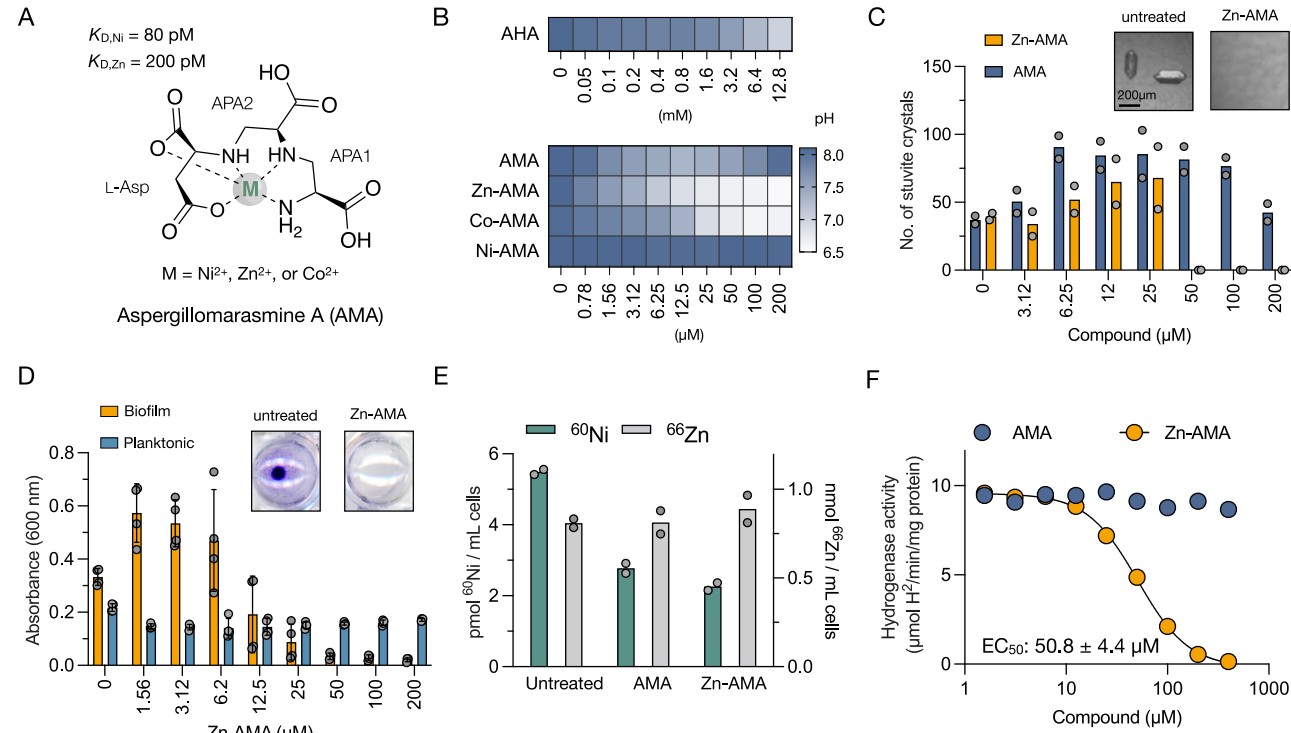

**Fig. 2 | Metal complexes of AMA suppress the activity of Ni-dependent enzymes. A** The metal complexed structure, selectivity, and affinity of AMA. The constituents of AMA are labeled L-aspartate (L-Asp) and 2,3-diaminopropionic acid (APA). **B** Dose-response urease inhibition assays in *K. pneumoniae*. Urease-dependent pH changes were determined after growth for 24 h at 37 °C in artificial urine. pH is represented by the color scale determined with phenol red. **C** Dose-response effect of AMA and Zn-AMA on the nucleation of struvite crystals in *K. pneumoniae* cultures grown in artificial urine. Crystals were examined microscopically in 96-well plates and visually quantified. The inset shows representative images of struvite crystals in the presence and absence of Zn-AMA. The data represent the representative mean of two replicates. **D** Quantification of planktonic

growth and crystal violet-stained biofilms of *K. pneumoniae* grown in artificial urine and varying concentrations of Zn-AMA. The inset shows representative images of crystal-violet-stained biofilms. Data represent mean ± s.d. of four technical replicates. **E** ICP-MS quantification of cellular $^{60}$Ni and $^{66}$Zn content in *K. pneumoniae* untreated or treated with 200 μM AMA, or Zn-AMA. Data represent the mean of two biological replicates (**F**) Dose-response curve of [NiFe]-hydrogenase inhibition in *K. pneumoniae* grown in LB-glucose for 24 h at 37 °C under anaerobic conditions. Whole-cell hydrogenase assays were performed by adding benzyl viologen (BV) at a final concentration of 1 mg/mL and monitoring its reduction at an absorbance of 630 nm. Data represent the mean of two technical replicates.

which validated components of the molecule that are important for NikA interactions.

# Results

## AMA binds Ni²⁺ with picomolar affinity

Our previous structural analysis of AMA determined that it binds to Ni²⁺ as a pentadentate complex with distorted octahedral geometry[28,31] (Fig. 2A). Based on this observation, we hypothesized that AMA might sequester Ni²⁺ outside bacteria and compete with its uptake. To further explore this possibility, we estimated AMA's $K_D$ for Ni²⁺ by competition binding between AMA, Zn²⁺, and Ni²⁺ using isothermal titration calorimetry (ITC). This approach was used because the $K_D$ for Ni²⁺ is outside the detection limit of direct titrations using ITC. The $K_D$ value for Ni²⁺ was estimated to be 80 pM, twofold lower than AMA's $K_D$ for Zn²⁺, and is consistent with the Irving-Williams order of ligand strength for these metals (Fig. 2A and Fig. S1). Compared to L-His, which forms a 2:1 complex with Ni²⁺ with $K_{D1}$ and $K_{D2}$ values of 0.64 μM and 8.8 μM at pH 7.5[34], AMA is a better nickelophore. Thus, it was anticipated that AMA would outcompete L-His for this metal in bacterial cultures, where the concentration of Ni²⁺ is typically sub-micromolar[35].

## AMA in complex with Zn²⁺ or Co²⁺ suppresses the activity of Ni-dependent enzymes by targeting Ni²⁺ uptake

Biological dose-response assays were performed to examine the effect of AMA on Ni²⁺ uptake. We used whole-cell urease-dependent pH changes as a readout to estimate cytoplasmic Ni²⁺ accumulation.

Experiments were conducted using urease-producing *K. pneumoniae* and *S. aureus* grown in artificial urine[36], each supplemented with varying concentrations of AMA or the clinically approved urease inhibitor acetohydroxamic acid (AHA)[37,38]. Varying concentrations of complexes of AMA with Zn²⁺, Co²⁺, or Ni²⁺ were also included as negative controls.

Contrary to our initial hypothesis, the data revealed that AMA did not significantly affect urease activity in *K. pneumoniae* (Fig. 2B and Fig. S2). However, its complexes with Zn²⁺ (Zn-AMA) and Co²⁺ (Co-AMA) significantly suppressed urease activity at a concentration of 50 μM. This suppression was notably more effective than AHA, which required a concentration of 12.8 mM to achieve similar results (Fig. 2B and Fig. S2). A contrasting effect was observed in *S. aureus*, where AMA, Zn-AMA, and Co-AMA inhibited urease activity similarly (Fig S2). This suggests that the AMA and its metal complexes inhibit urease activity through multiple mechanisms in *S. aureus*. Nevertheless, AMA in complex with Ni²⁺ (Ni-AMA) or ZnCl₂ did not affect urease activity in either organism (Fig. 2B, S2 and S3). Given these observations, spot dilutions were performed on cells treated with 200 μM of AMA, Zn-AMA, Co-AMA, and Ni-AMA to exclude the possibility of urease attenuation being due to lethal effects on the bacteria. The results showed that these compounds do not affect bacterial viability after 24 h of exposure (Fig S4).

We next examined artificial urine cultures of *K. pneumoniae* microscopically to determine if urease-dependent struvite crystallization had been affected. While Zn-AMA robustly prevented the

appearance of struvite crystals, AMA did not, which was consistent with our observations in whole-cell urease assays (Fig. 2C and Fig. S5). However, at low compound concentrations, both resulted in a marginal pH-independent increase in crystal nucleation, which could not be accounted for in our experimental conditions.

Given that struvite crystallization could be inhibited, we evaluated whether Zn-AMA could suppress the formation of biofilms, as bacteria can attach to struvite crystals and embed them in extracellular polysaccharides[39]. In a microtiter plate biofilm assay, Zn-AMA could completely suppress biofilm production in *K. pneumoniae* culture grown in artificial urine, tracking with the anti-urease/struvite activity of the compound (Fig. 2D and Fig. S6). This effect was specific to urine-like conditions, as biofilms were unperturbed by the compound when bacteria were grown in M9 minimal medium. Such anti-biofilm activity was not observed in *S. aureus* cultures grown in either condition, highlighting differences in the biology of its biofilms compared to *K. pneumoniae* (Fig S6).

To guide our exploration of the mode of action of Zn-AMA, we investigated whether a chemical-chemical interaction occurs with the urease inhibitor AHA. We hypothesized that a synergistic effect could be observed with AHA if Zn-AMA inhibited urease activity indirectly. This could occur through disruption of urease maturation in *K. pneumoniae*, specifically by interfering with the uptake of $Ni^{2+}$. Conversely, an additive effect was anticipated if these compounds had equivalent or overlapping mechanisms. We focused on the $Zn^{2+}$ complex because of the potentially toxic effects of $Co^{2+}$. Using a checkerboard assay, we found the combination of Zn-AMA and AHA was synergistic, indicating that Zn-AMA affects urease indirectly (Fig. S7). This was further verified with a cell-free urease assay, which excluded the direct inhibition of urease by AMA and Zn-AMA (Fig. S7). We subsequently performed inductively coupled plasma mass spectrometry (ICP-MS) analyses to evaluate the intracellular $Ni^{2+}$ and $Zn^{2+}$ content of *K. pneumoniae*. The data showed that AMA and Zn-AMA similarly reduced intracellular $Ni^{2+}$ levels without affecting $Zn^{2+}$ levels, indicating that Zn-AMA has pleiotropic effects on $Ni^{2+}$ homeostasis. (Fig. 2E).

We reasoned that disruption of $Ni^{2+}$ uptake and homeostasis should also perturb the activity of other Ni-dependent enzymes. To examine this possibility, we tested the effects of free AMA and Zn-AMA on Ni-dependent hydrogen metabolism in bacteria. We focused on *K. pneumoniae* because *S. aureus* does not possess [NiFe] hydrogenases based on a search in HydDB[21]. *K. pneumoniae* produces a single [NiFe]-hydrogenase (Hyd-3) as part of the $H_2$-evolving formate hydrogenlyase (FHL) complex and is encoded by the *hyc* operon (Fig S8)[21]. Dose-response assays for [NiFe]-hydrogenase inhibition were performed on *K. pneumoniae* grown in LB-glucose under anaerobic conditions (10% $CO_2$, 5% $H_2$, 85% $N_2$). To measure [NiFe]-hydrogenase-catalyzed $H_2$ reduction, benzyl viologen was used as an artificial electron acceptor, as previously described[40]. Although Hyd-3 is involved in the evolution of $H_2$, previous work in *E. coli* has shown that it is responsible for 90% of the hydrogen-dependent benzyl viologen reduction in crude extracts and whole-cell assays[41–44]. Our results revealed that Zn-AMA inhibited *K. pneumoniae* Hyd-3 activity in a dose-dependent manner with an $EC_{50}$ value of $50.8 \pm 4.4\,\mu M$ (Fig. 2F). Notably, free AMA had no impact on [NiFe]-hydrogenase activity, further highlighting the unique qualities of Zn-AMA. The distinct effects of these compounds on [NiFe]-hydrogenase activity were verified with zymographic analysis of Hyd-3 in *E. coli* cell lysates, which can be differentiated by exposure to 100% $H_2$ (Fig S9)[42].

## Metal complexes of AMA target NikA of the Nickel uptake system

Our observation that Zn-AMA reduces total cellular $Ni^{2+}$ content led us to hypothesize that AMA competes with $Ni^{2+}$ uptake through the NikABCDE ABC transporter. Previous work has shown that the periplasmic solute binding protein NikA binds to non-native metal complexes and that $\Delta nikA$ strains lack Ni-dependent enzyme activity[26,45]. We tested this hypothesis by titrating Ni-(L-His)$_2$ into cells containing varying amounts of Zn-AMA and observed that Ni-(L-His)$_2$ rescued urease activity in a dose-dependent manner, consistent with the idea that both complexes bind to the same receptor (Fig. 3A).

To further examine this common connection through NikA, *K. pneumoniae* was grown in a minimal medium containing urea as the sole nitrogen source. Under this condition, cellular viability becomes $Ni^{2+}$-dependent because urease activity is necessary for metabolizing urea to $NH_3$ for nitrogen assimilation[20]. Therefore, we hypothesized that if metal complexes of AMA could bind to NikA, then Ni-AMA may rescue growth under these conditions. Growth curves showed that cells were not viable when supplemented with L-His or AMA alone (Fig. 3B, C). However, Ni-(L-His)$_2$ and Ni-AMA complexes stimulated growth (Fig. 3B, C), while Zn-AMA or nickel-selective chelator dimethylglyoxime (DMG) did not (Fig. 3C). Indeed, Zn-AMA inhibited the growth of *K. pneumoniae* in this medium when supplemented with $0.25\,\mu M$ NiCl, which was suppressed with the addition of excess Ni-(L-His)$_2$ (Fig. S10). These results indicate that cells likely transport AMA complexes into the cell through the NikA pathway, while Ni-DMG sequesters $Ni^{2+}$ and prevents its uptake and utilization.

We validated NikA as the target of AMA metal complexes with cellular thermal shift assays (CETSA)[46]. NikA levels were monitored in *K. pneumoniae* with immunodetection of a FLAG-tagged form of the protein and OmpA as a loading control. CESTAs showed that NikA levels began to decrease at 52.8 °C and became barely detectable at 60 °C in both AMA-treated and untreated control cells (Fig. 3D and Fig. S11). Ni-AMA increased the thermal stability of NikA, which remained detectable at 60 °C (Fig. 3D and Fig. S11) and in vitro thermal shift assays using recombinant NikA confirmed that Ni-AMA, Zn-AMA, and Co-AMA increase NikA's stability (Fig. 3E). In experiments where only the metals were added, $Zn^{2+}$ stabilized NikA, while $Ni^{2+}$ and $Co^{2+}$ did not affect it (Fig. 3F). This $Zn^{2+}$-dependent increase in thermal stability was not observed in the Ni-AMA bound form of NikA, suggesting that the binding of these reagents is mutually exclusive (Fig. S12). The biological significance of $Zn^{2+}$ binding by NikA remains unclear; however, it is widely acknowledged in the literature that $Zn^{2+}$ has the potential to affect the stability and folding dynamics of proteins, regardless of the biological relevance of the metal interaction[47–49].

## NikA binds to metal complexes of AMA with high affinity

Since metal complexes of AMA bind NikA, their binding affinities and thermodynamics were determined using ITC. To set the benchmark for high-affinity ligand interactions with *K. pneumoniae* NikA, the known ligand Ni-(L-His)$_2$ was titrated into a solution of purified protein. The binding of Ni-(L-His)$_2$ was exothermic with a 1:1 stoichiometry, and binding/thermodynamic parameters were analogous to the NikA homolog from *E. coli*[50] (Fig. 4A). The titration of Ni-AMA was similarly exothermic, enthalpy-driven, and had a $K_D$ value of $2\,\mu M$, which was 4-fold greater than Ni-(L-His)$_2$ (Fig. 4B). Comparable thermodynamics were obtained for Zn-AMA and Co-AMA (Fig. 4C, D). However, incrementally increasing $\Delta H°$ values were observed for Zn-AMA and Co-AMA, respectively. These $\Delta H°$ increases corresponded with increases in the $K_D$ values for Zn-AMA and Co-AMA, which shifted 3.5 and 7-fold relative to Ni-AMA. Even though these complexes have different affinities for NikA, we predict that their structures are pentadentate with octahedral geometry, given the known structural/conformational restraints observed in the Ni-AMA structure[31]. Therefore, the affinity differences are likely caused by distinct metal-ligand distances in each complex, impacting the fit within the binding pocket of NikA. Alternatively, water could serve as a $6^{th}$ ligand and affect each complex's affinity for NikA, but the stability of this interaction for each complex is unknown. Despite the minor differences in affinity among the various AMA metal complexes, NikA's affinity for

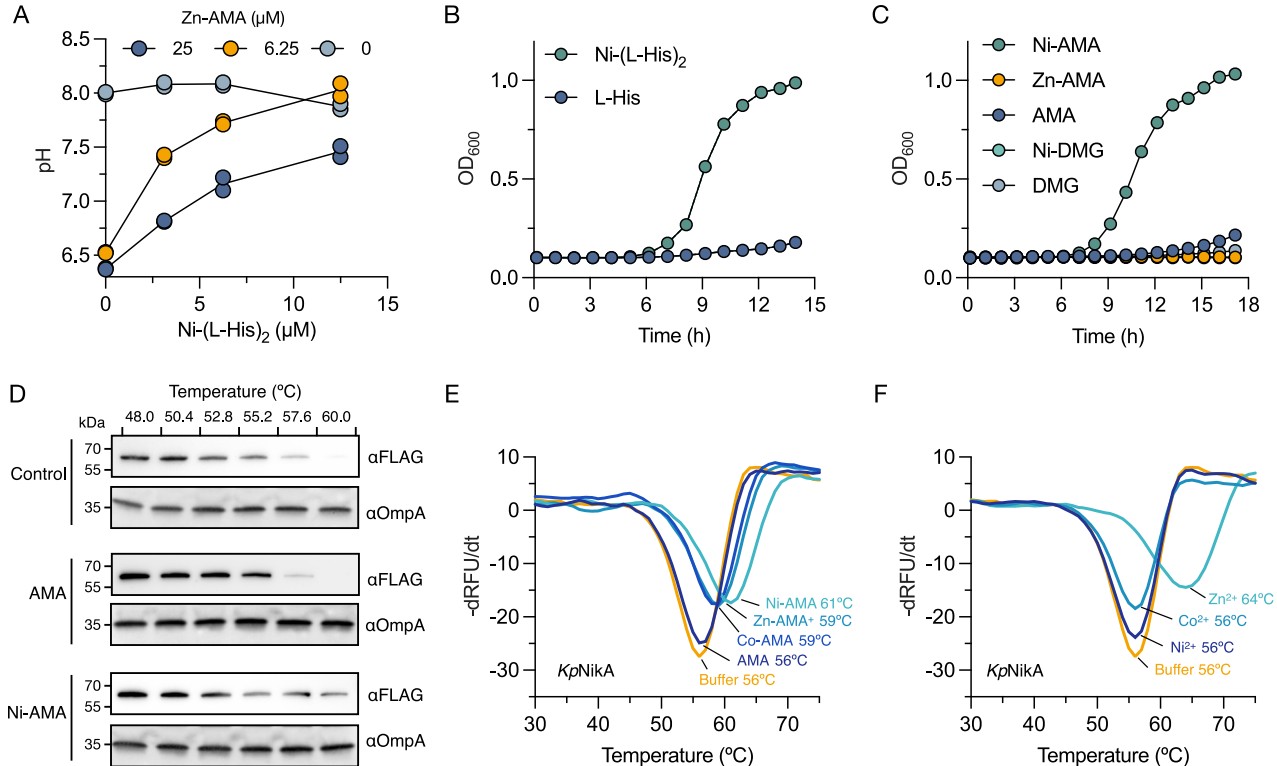

**Fig. 3 | Bacterial nickel uptake is inhibited by competitive binding to the periplasmic Ni-binding protein NikA from *K. pneumoinae*. A** Zn-AMA complexes compete for uptake with the nickelophore Ni-(L-His)$_2$. *K. pneumoniae* was cultured in artificial urine for 24 h at 37 °C with varying amounts of Zn-AMA and Ni-(L-His)$_2$ and the pH of the culture was determined with phenol red. The data represent the mean of duplicate experiments. **B**, **C** Growth kinetics of *K. pneumoniae* in Chelex-100 treated nitrogen-limited media containing urea as a sole nitrogen source supplied with Ni-(L-His)$_2$ (25 µM), L-His (25 µM), Ni-AMA (50 µM), Zn-AMA (50 µM), AMA (50 µM), Ni-DMG (50 µM), or DMG. Data represent the mean of duplicate experiments. **D** Cellular thermal shift assays (CETSA) of *K. pneumoniae* producing NikA-FLAG. Relative to the untreated control, AMA (250 µM) and Ni-AMA (250 µM) were incubated with cells for 30 min, followed by heating over a gradient of temperatures. Cell lysates were analyzed by immunoblotting with anti-FLAG-HRP (αFLAG) and anti-OmpA (αOmpA) antibodies. Immunoblots are representative of two independent experiments. **E**, **F** Negative derivative melt plots using purified recombinant NikA from *K. pneumoniae*. Metals or complexes (100 µM) were added to NikA (1 µM) with 2× SYPRO orange. The *y* axis is displayed as negative derivative (−dRFU/dt) plots of the raw output.

Zn-AMA is 14-fold weaker than Ni-(L-His)$_2$. Given the relatively low concentration of Ni$^{2+}$ in the environment, it is reasonable to expect that Zn-AMA interacts with NikA strongly enough to outcompete Ni-(L -His)$_2$ in living bacteria.

## Crystal structure of NikA in complex with Ni-AMA

*E. coli* NikA (22-524), which shares 75% identity and 90.5% similarity with the *K. pneumoniae* ortholog, was purified for crystallization studies. The structure of the NikA was determined in complex with Ni-AMA to 2.15 Å resolution by molecular replacement using the Ni-(L-His)$_2$ (PDB 4I8C) as a search model (Fig. 5A) (Table S1), and Ni-AMA could be confidently placed within the electron density map (Fig. 5B). Ni-AMA adopted different coordination geometry when bound to NikA compared to the stand-alone structure determined in our previous work[31], indicating that two configurations of the Ni-AMA complex are possible. Superposition of the Ni-AMA structures showed that relative to AMA's L-Asp unit (L-Asp$^{AMA}$), the penultimate APA2 and N-terminal APA1 are rotated by 93.3° (Fig. 5C). Consequently, the NH$_2$ ligand of the terminal APA1 subunit is equatorial instead of axial as seen in the stand-alone Ni-AMA structure.

The interaction network between NikA and Ni-AMA revealed that the conserved His416 residue occupies the axial position within the Ni-AMA complex (Fig. 5A, D). This observation suggests that NikA either conformationally selects for one form of Ni-AMA or that Ni-AMA binding to NikA induces a conformational change to accommodate the Ni$^{2+}$ interaction with His416. The remainder of the ligand binding pocket of NikA around the Ni-AMA is composed of Tyr22,

Met27, Trp100, Tyr382, Trp398, and Tyr402 (Fig. 5D). Notable CH-π stacking interactions occur between Trp100 and APA2, and APA1 and Tyr382. Extensive electrostatic interactions also occur with the carboxyl groups of Ni-AMA, including hydrogen bonds with the side chain of Arg137 to the α-carboxyl of the L-Asp$^{AMA}$ residue. The terminal APA1 carboxyl makes an additional hydrogen binding to the backbone amide of Thr23. Ser415 and Arg97 form hydrogen bonds to the carboxyl group of APA2 (Fig. 5D). Several other hydrogen bond interactions occur between Ni-AMA and ordered solvent molecules within the pocket. Together, these results provide the structural basis for the high-affinity binding of Ni-AMA, serving as a proxy for the binding of complexes composed of other metal ions.

## The Ni-AMA complex mimics Ni-(L-His)$_2$

Ni-AMA induced the closed conformation of NikA (Fig. 6A)[51]. Only a small opening between the two lobes remains, slightly exposing the APA1 carboxyl group. Structural comparisons of the closed form of the NikA:Ni-AMA complex to that of the NikA:Ni-(L-His)$_2$ complex (PDB 4I8C) revealed an overall RMSD of 1.16 Å$^2$ (over 496 residues). Superpositions of lobe-I revealed a subtle difference in the position of lobe-II, which contains His416 that interacts with Ni-AMA. Unlike the NikA:Ni-AMA structure, the binding pocket of the Ni-(L-His)$_2$ complex remains ajar to accommodate the large side chain of the L-His(B) residue, which sterically affects the position of the α11 helix of lobe II by occluding the side chain of Tyr382. An additional hydrogen bond to Arg386 stabilizes the position of this alpha-helix through a hydrogen

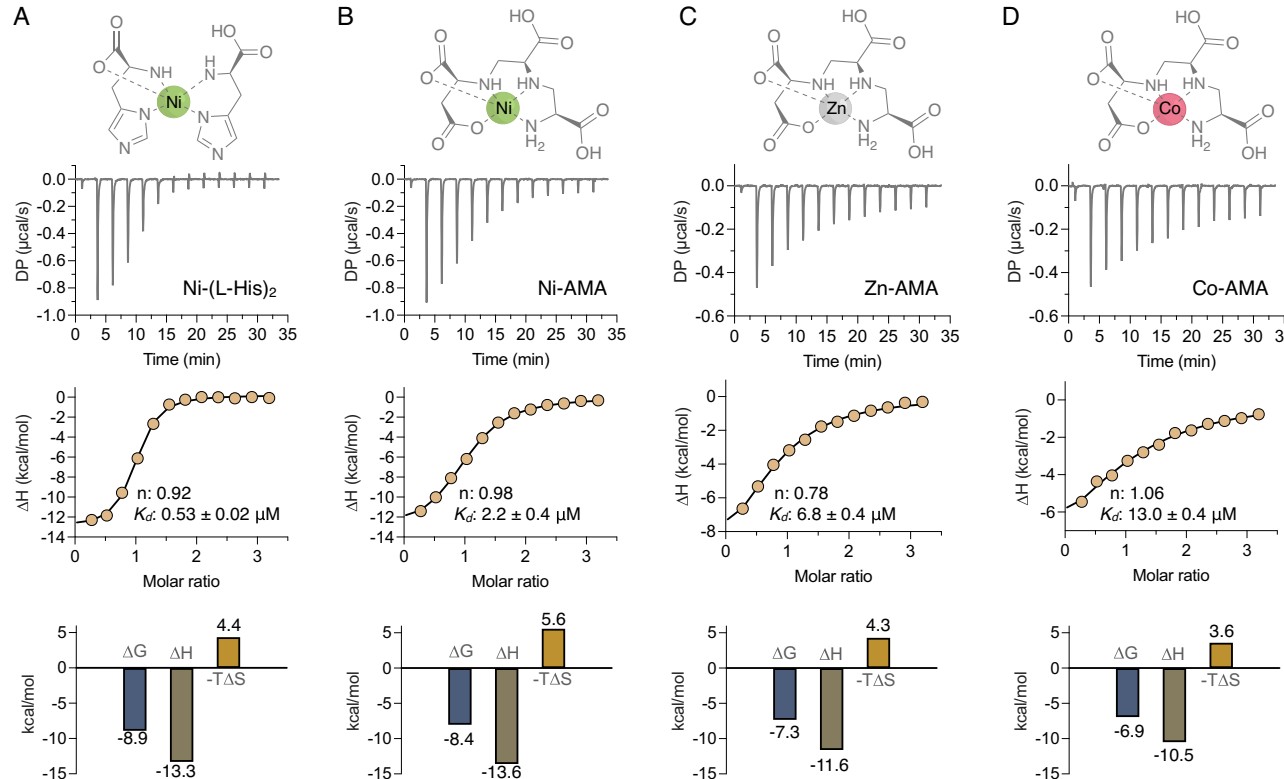

**Fig. 4 | Metal complexes of AMA bind to NikA with high affinity. A–D** Isothermal titration calorimetry analyses of the interactions between various metal complexes and NikA from *K. pneumoniae*. The metal complexes (250 µM), prepared at a 1:1 molar ratio (AMA) or 2:1 molar ratio (L-His), were titrated into NikA (15 µM) in 25 mM Tris-HCl, 150 mM NaCl, 0.1% (v/v) Triton X-100, pH 7.5 at 37 °C. Top, raw titration; middle, Wiseman isotherm of integrated peaks; bottom, binding signatures of free energy (ΔG), enthalpy (ΔH), and entropy (−TΔS). Ni-(L-His)₂ and Ni-AMA structures represent the NikA-bound form (PDB 4I8C) and free complex, respectively. Zn-AMA and Co-AMA structures are predicted based on the known structure of Ni-AMA.

bond to the carboxyl group L-His(B). (Fig. 6B, D). In contrast, Ni-AMA lacks this bulky side chain at the equivalent position, allowing for a fully closed pocket. (Fig. 6B, D). Further comparisons to other NikA structures available in the PDB identified that a NikA:Ni-butane1,2,4-tricarboxylate (BTC) complex (PDB 3DP8) and a NikA:Fe-EDTA complex (PDB 1ZLQ) have RMSD values of 0.46 Å and 1.34 Å, respectively (over 496 residues). Like Ni-AMA, Ni-BTC also lacks a bulky group that would otherwise block the complete closure of the pocket (Fig. 6C). In contrast, the NikA-Fe-EDTA structure is in a partially open state. With Fe-EDTA being a hexadentate ligand, it sterically hinders the complete closure of the ligand-binding site. Comparing all four ligands revealed that the AMA, L-His(A), BTC, and EDTA occupy a similar region within the binding pocket, overlapping with L-Asp^(AMA) of Ni-AMA. The structure of the NikA:Ni-AMA complex shows that AMA metal complexes are reasonable Ni-(L-His)₂ mimetics and that full NikA closure and high-affinity binding are likely facilitated by accepting His416 as the 6th Ni²⁺ ligand (Fig. 6C). This is on contrast to Fe-EDTA, which does not contain an available coordination site, which precludes engagement with His416. (Fig. 6C). These structural insights into the plasticity of NikA's binding pocket will inform the selection of AMA analogs for structure-activity relationship studies.

## Structure-activity relationship studies

To study the structure-activity relationships of AMA, we tested synthetic and natural analogs with substitutions at APA1 and L-Asp^(AMA) positions we prepared previously[31]. We prioritized testing Zn²⁺ complexes instead of Co²⁺ and Ni²⁺ complexes due to potential toxicity and because Ni-AMA does not reduce urease activity.

The side chain of L-Asp^(AMA) is adjacent to the aromatic side chain of Trp398 within the crystal structure of the NikA complex (Fig. 5D).

To potentially improve this interaction, L-Asp^(AMA) was substituted with L-His^(AMA) and L-Asn^(AMA), capitalizing on potential π-π and amide-π stacking interactions. ITC experiments revealed that these substitutions improved the affinity for NikA 7-fold, as the $K_D$ values for both compounds were ~1 µM (Table 1 and Fig. S13). As anticipated, these improvements correspond to a significant decrease in the ΔH values, indicating increased electrostatic interactions. Despite the increased affinity, the minimum inhibitory concentrations for urease activity were uncorrelated as the EC₅₀ for Zn-AMA(His) and Zn-AMA(Asn) was >200 µM and 10-fold greater than Zn-AMA, respectively (Table 1 and Fig. S14).

On the opposite side of AMA, APA1 forms several hydrogen bonds and is braced by a CH-π interaction with the aromatic side chain of Tyr382 (Fig. 5D). To understand the importance of APA1; we tested toxin A, an AMA analog that lacks this component, and aspergillomarasmine B (AMB), which contains a glycine residue at this position instead. The $K_D$ of Zn-toxin A was similar to AMA but exhibited different binding thermodynamics (Table 1 and Fig. S13). The thermodynamic differences suggest less conformational restriction of Zn-toxin A occurs upon binding relative to Zn-AMA. Despite having low micromolar affinity, Zn-toxin A was found to have poor biological activity (EC₅₀ > 200 µM) (Table 1 and Fig. S14). In contrast, AMB had a 4-fold weaker affinity toward NikA characterized by less favorable ΔH from a lack of hydrogen bonding potential (Table 1 and Fig. S13). In keeping with the weak affinity of AMB, we observed an EC₅₀ of >200 µM in bacterial culture (Table 1 and Fig. S14). These data indicate that while NikA-AMA interactions can be improved, other factors significantly influence biological activity. We anticipate that it may be attributable to a combination of Zn²⁺ affinity, metal selectivity, and cellular accumulation.

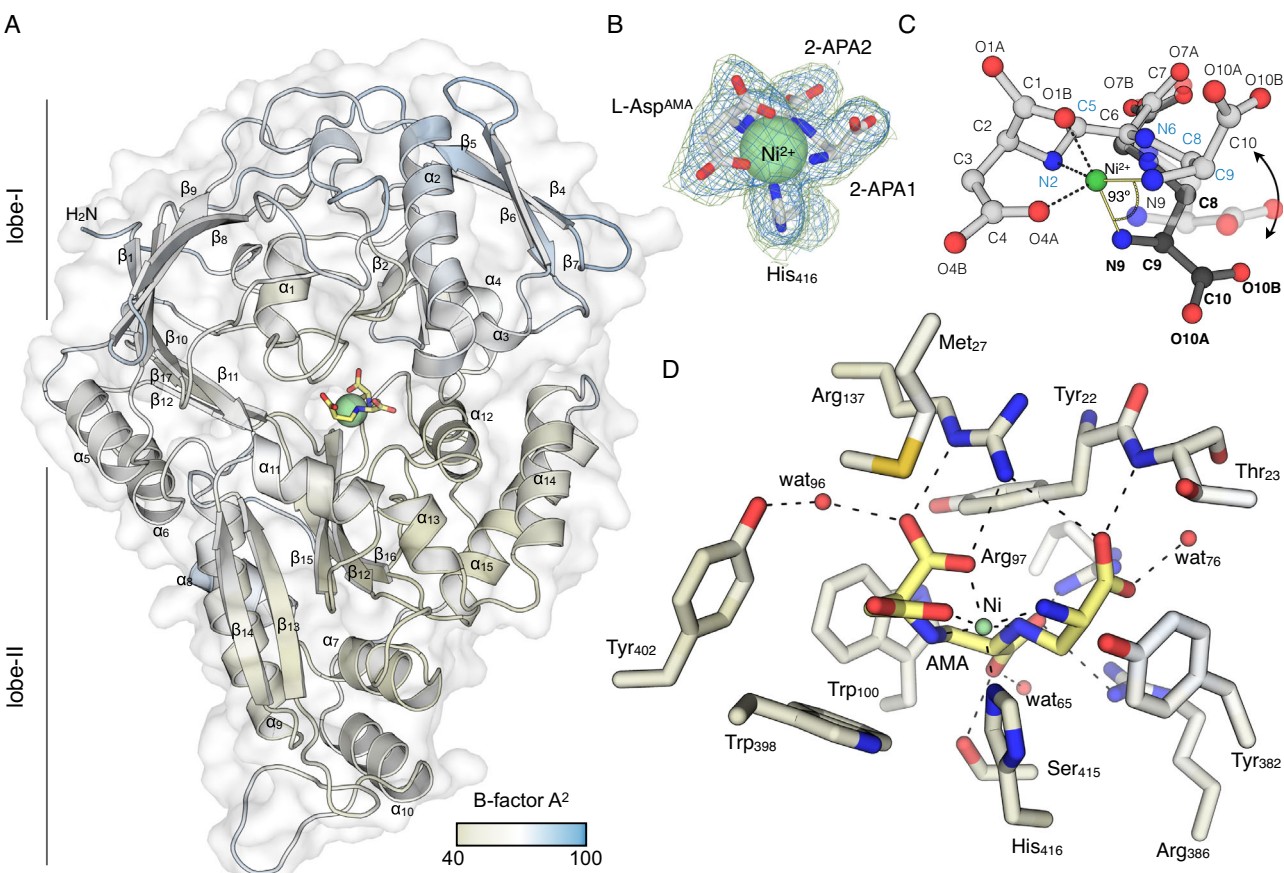

**Fig. 5 | Crystal structure of *E. coli* NikA in complex with Ni-AMA. A** Cartoon/ surface representation of NikA colored by B-factor. AMA and $Ni^{2+}$ are shown in stick and sphere representation, respectively. The color bar depicts the B-factor scale. **B** The $2mF_o\text{-}DF_c$ map and $mF_o\text{-}DF_c$ omit map are shown in blue and green mesh and contoured around the Ni-AMA complex and His416 at 1.0 σ and 3.0 σ, respectively. **C** Ball and stick structures of NikA-bound Ni-AMA (gray) and free Ni-AMA (black) are shown superposed and depict the possible intermediate between the two states (transparent). Atomic labels corresponding to NikA-bound AMA and free AMA are shown in light and bold font, respectively. **D** Detailed interactions between the residues of NikA's ligand binding pocket within 5 Å of Ni-AMA are shown in stick representation. NikA residues are colored by B-factor according to the color bar in (A), with AMA colored in yellow. Non-bonded solvent atoms (red) and $Ni^{2+}$ (green) are depicted as spheres. All possible hydrogen bonds are shown as black dashed lines. In all panels, nitrogen and oxygen atoms are colored blue and red, respectively.

## In vivo toxicity and efficacy studies

We next assessed the cytotoxicity of Zn-AMA and compared it to AMA, Co-AMA, and Ni-AMA toward human embryonic kidney 293 (HEK293) cells. The cells were treated with varying amounts of each compound, with the highest test concentration fixed at 512 μg/mL (-1.4 mM). Zn-AMA, AMA, and Ni-AMA showed no significant cytotoxicity over 48 h (Fig. S15). In contrast, Co-AMA resulted in a 50% reduction in HEK293 cell viability at the highest test concentration, indicating that it has some cytotoxicity (Fig. S15).

Since Zn-AMA was not toxic in vitro, we tested its efficacy in vivo using *Galleria mellonella* larvae, serving as a model for the innate immune response. Although Zn-AMA does not affect bacterial viability in vitro, the in vivo efficacy data showed that Zn-AMA was an effective monotherapy for both *K. pneumoniae* and a methicillin-resistant strain of *S. aureus* (USA300) at 7.5 mg/kg (Fig. 7AB). A significant proportion of the *G. mellonella* larvae survived following infection by both species after seven days post-infection relative to the untreated control. This was consistent with the improved survival of an *S. aureus* USA300 *ureC::tn* transposon mutant lacking urease activity (Fig. 7C). Moreover, treating the *ureC::tn* mutant with Zn-AMA showed no significant survival benefit, indicating that the efficacy of Zn-AMA is predominantly associated with urease attenuation in vivo (Fig. 7C). In summary, the findings suggest that using Zn-AMA to inhibit $Ni^{2+}$ uptake is an effective strategy for reducing the virulence of *K. pneumoniae* and *S. aureus* when used as a single treatment.

## Discussion

The significance of urease in medicine and agriculture has sustained considerable scientific, economic, and environmental interest. In medicine, ureolysis by urease-producing bacteria was recognized as the cause of struvite stones in the 1940s[52] and in the mid-1980s[53], it was identified as a critical virulence determinant of *H. pylori*. These discoveries, among many others, established that urease is an important target for non-traditional antimicrobial chemotherapy in infections caused by urease-producing organisms. In 1980, AHA (sold as Lithostat) was approved to combat struvite formation in humans[37] but has remained underutilized because of concerns over safety due to its acute toxicity and carcinogenicity[38,54]. The search for urease inhibitors has since continued, yielding countless compounds from natural and synthetic sources, but a promising lead has yet to justify further development (-3200 molecules as of 2022)[55]. As an alternative approach to direct urease inhibition, a recent investigation of the mode of action of colloidal bismuth citrate identified that it perturbs $Ni^{2+}$ trafficking to block the assembly of the di-$Ni^{2+}$ metallocentre in urease to indirectly inhibit ureolysis[56].

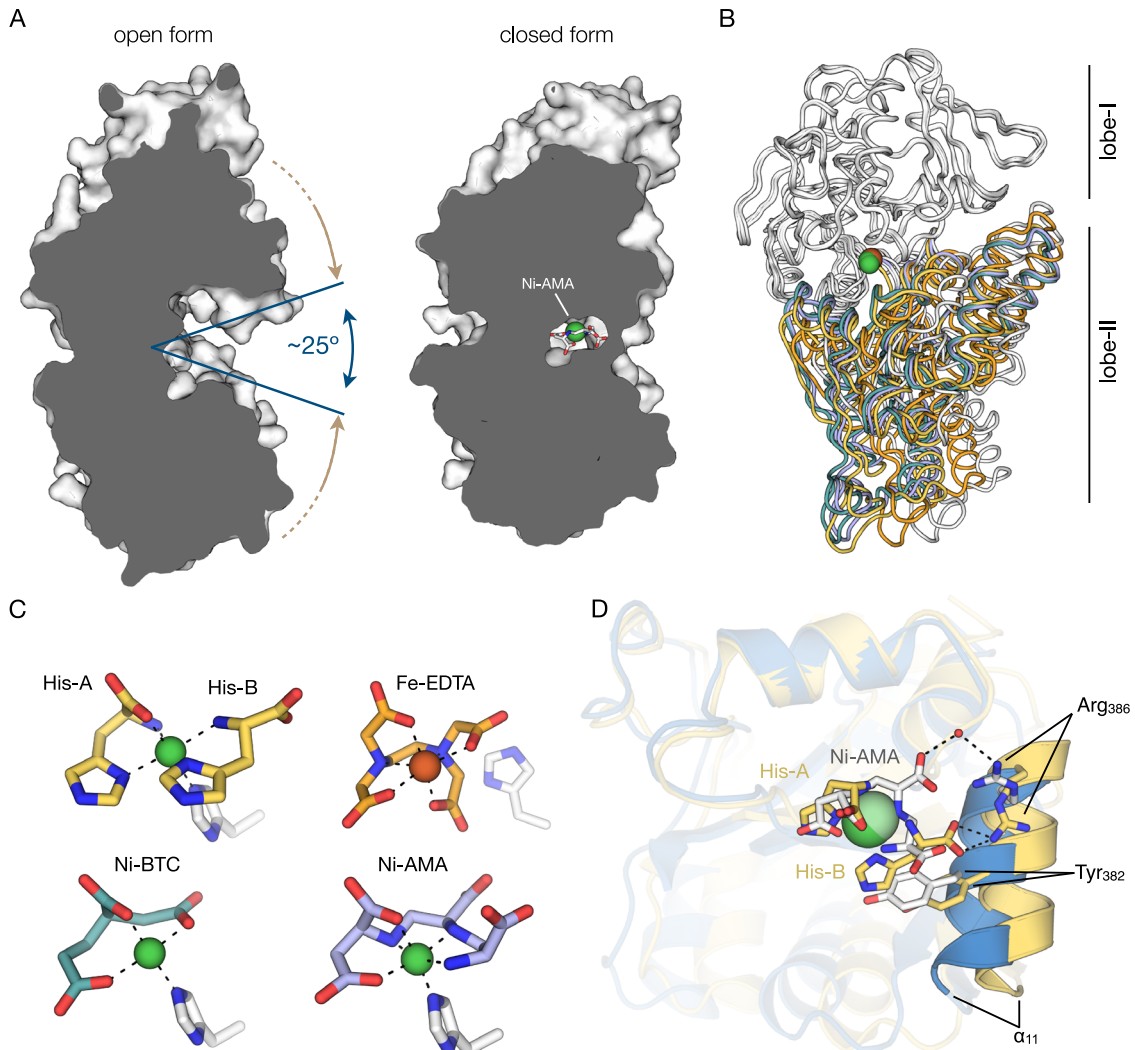

**Fig. 6 | Structural comparison of NikA:Ni-AMA to known ligand-bound structures of NikA. A** Surface representation of NikA apoprotein in its open form (PDB 4UIU) compared to the Ni-AMA bound form of NikA. AMA and $Ni^{2+}$ are shown as sticks (gray) and spheres (green). **B** Structural superposition of NikA:Ni-AMA (light blue), NikA:Ni-(L-His)$_2$ (PDB ID 4I8C; yellow), NikA:Ni-BTC (PDB ID 3DP8; teal), NikA:Fe-EDTA (PDB 1ZLQ; orange), and NikA apoprotein (PDB 1UIU; white) aligned relative to lobe-I (gray) shown in ribbon representation. **C** Structural comparison of $Ni^{2+}$ complexes from **B**. Complexes are colored according to the scheme from **B**, except His416, which is shown as white sticks. **D** Cartoon detail of the lobe-II overlay from **B** showing NikA:Ni-AMA (blue) and NikA:Ni-(L-His)$_2$ (yellow). Ni-AMA (white) and Ni-(L-His)$_2$ (yellow) are shown in stick representation.

While urease inhibition has been extensively explored, emerging interest in inhibiting [NiFe]-hydrogenases presents new opportunities and challenges in developing antimicrobial therapies[33,41,57,58]. Advantages to inhibiting [NiFe]-hydrogenases include being absent from humans and reducing bacterial virulence and viability. However, a potential challenge with inhibitor development is the diversity of these enzymes in pathogenic bacteria. There are four distinct classes of [NiFe]-hydrogenase[21], each of which is not equally represented in bacteria, suggesting a single compound may not be equally effective toward all classes and pathogens. This need for a broad-spectrum inhibitor may pose significant difficulties in compound development, particularly in achieving high affinity, potency, and efficacy.

Our work on AMA has shown that blocking the uptake of environmental $Ni^{2+}$ is a viable approach to inhibit multiple Ni-dependent enzymes, including urease and [NiFe]-hydrogenases, to block urine alkalinization, struvite, and biofilm formation, and $H_2$ metabolism. It was shown previously that EDTA and the Ni-specific chelator DMG could similarly prevent $Ni^{2+}$ uptake by sequestering the metal outside the cell[59,60]. In contrast, we show that complexes of AMA with $Zn^{2+}$ or $Co^{2+}$ specifically compete for $Ni^{2+}$ uptake by binding to NikA. A benefit to using preformed metal complexes of AMA compared to a chelator-based approach is offsetting the potentially adverse off-target effects of metal chelation. One potential drawback, however, is the stimulatory effect of Ni-AMA on urease activity. Given its affinity for AMA, $Ni^{2+}$ could displace $Zn^{2+}$ and aid in microbial uptake; however, this is unexpected given the low abundance of $Ni^{2+}$ in most biological fluids (<5 µg/mL; 90 nM)[61].

Ni-(L-His)$_2$ has been identified as a common ligand of NikA homologs and has been co-crystallized with several Ni-binding proteins, including orthologs from *Campylobacter jejuni, Yersinia pestis, H. pylori, S. aureus,* and *E. coli*. Our observation that the three-dimensional structure of Ni-AMA closely mimics Ni-(L-His)$_2$ provides a structural basis for targeting Zn-AMA to diverse bacteria. The polycarboxylic acid structure of AMA enables the formation of an extensive hydrogen bond and π-interaction network within NikA's binding site. Furthermore, the pentadentate coordination geometry of AMA offers the flexibility of both high affinity binding to $Zn^{2+}$ while providing an unoccupied coordination site for interactions with the His416 ligand of NikA.

**Table 1 | Structure-activity relationships of AMA analogs in complex with Zn²⁺**

| Structure | Compound | EC₅₀ᵃ (µM) | $K_D^b$ (µM) | ΔG (kcal/mol) | ΔH (kcal/mol) | −TΔS (kcal/mol) |
|---|---|---|---|---|---|---|
|  | Zn-AMA | 2.5 ± 0.4 | 6.8 ± 0.4 | −7.3 | −11.6 | 4.3 |
|  | Zn-AMA(Asn) | 24 ± 6 | 1.3 ± 0.1 | −8.4 | −13.1 | 4.7 |
|  | Zn-AMA(His) | >200 | 0.8 ± 0.1 | −8.6 | −13.1 | 4.5 |
|  | Zn-AMB | >200 | 24 ± 5 | −6.5 | −9.7 | 3.1 |
|  | Zn-Toxin A | >200 | 6 ± 0.8 | −7.4 | −9.6 | 2.2 |

ᵃHalf-maximal effective concentration required to suppress urease-dependent pH change.
ᵇMean ± s.e. of three replicate experiments.

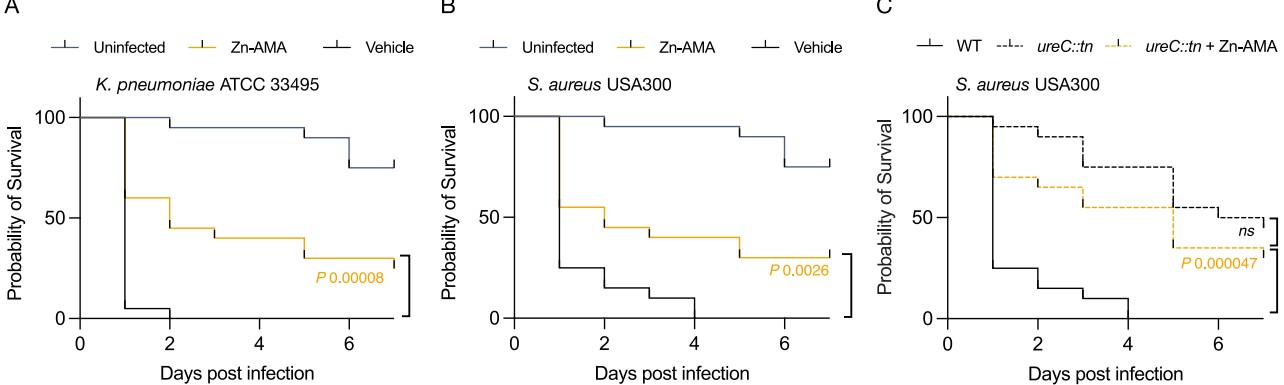

**Fig. 7 | Zn-AMA attenuates virulence in vivo.** The probability of *G. mellonella* larvae survival infected with (**A**) *K. pneumoniae* ATCC 33495, or (**B**) *S. aureus* USA300, treated with vehicle (PBS; blue) or Zn-AMA (7.5 mg/kg; yellow), compared to the uninfected control injected with PBS (black). The results were obtained from two groups (ten larvae per group) representing the mean. **C** The probability of *G. mellonella* larvae survival infected with *S. aureus* USA300 *ureC::tn*, treated with vehicle (PBS; blue), or Zn-AMA (7.5 mg/kg; yellow). The results were obtained from two groups (ten larvae per group) representing the mean. The *P* value indicates treatments where survival was statistically significant to the vehicle (Log-rank test). *ns* shows non-significant differences.

Structure-activity relationship studies indicate that ʟ-His and ʟ-Asn analogs of AMA can improve the overall interaction with the binding site and generate high-affinity values comparable to Ni-(ʟ-His)₂. Downstream of NikA binding, the fate of AMA/analog complexes is unclear, but we envisage three possible outcomes. (i) The entire complex is imported into the cytoplasm through the NikBCDE transporter as an inert metallophore. (ii) Zn²⁺ is released from AMA by NikA at the interface of the ABC transporter, allowing only the metal to pass through, which could interfere with enzyme maturation. (iii) The conformation of NikA in complex with Zn-AMA and analogs is incompatible with the NikBCDE transporter, and the complex remains bound to the protein in the periplasm. Considering the molecular complexity of the uptake system and that little is known about how the NikABCDE transporter translocates cognate Ni²⁺ complexes, future studies will investigate the biochemical outcome of Zn-AMA on translocation to inform the development of additional analogs.

Antibiotic-resistant infections are a global health crisis, and new approaches are urgently needed to overcome them. In this work, we show that targeting NikA to block the uptake of Ni²⁺ in several human pathogens could be a feasible strategy to attenuate Ni-dependent virulence determinants of many bacteria. The versatility of metal complexes has recently gained attention as agents to confront

antimicrobial resistance, underscoring the significance of our findings[62]. Specifically, Zn-AMA represents a new example of a metal-based compound with therapeutic potential, setting a precedent for identifying additional analogs with improved biological activity and broad-spectrum activity.

## Methods

### Bacterial strains and growth conditions

*K. pneumoniae* ATCC 33495 and *S. aureus* ATCC 23293 were used to evaluate the effects of AMA. Where indicated, bacteria were grown in artificial urine, nitrogen-limiting minimal medium (10.5 g $K_2HPO_4$, 4.5 g $KH_2PO_4$, 2% (w/v) urea, 0.4% (w/v) glucose, 1 mM $MgSO_4$, 1 μM $ZnCl_2$, and 1 μM, $FeCl_3$) Chelex-100 treated to remove trace $Ni^{2+}$ before the addition of $Mg^{2+}$, $Zn^{2+}$, and $Fe^{3+}$, M9Cas (M9 minimal medium supplemented with 0.1% (w/v) casamino acids), LB-glucose (LB supplemented with 0.8% (w/v) glucose, 10 mM $MgSO_4$, 10 mM potassium phosphate). Artificial urine was prepared according to ref. 36 and supplemented with 0.1% casamino acids and 0.01% (w/v) phenol red. For the growth of *S. aureus*, artificial urine was supplemented with 10% (v/v) LB. *E. coli* TOP10 was used for constructing, maintaining, and propagating plasmid DNA, and *E. coli* BL21 (DE3) was used for protein production. Bacteria were routinely grown in liquid culture under aerobic conditions at 37 °C using either Luria-Bertani (LB; Lennox formulation) supplemented with 50 μg/mL kanamycin when appropriate. LB agar, supplemented with 50 μg/mL kanamycin when appropriate, was used for growth on a solid medium. All bacteria were grown at 37 °C.

### DNA manipulation

For protein production, plasmids containing truncated nikA homologs from *K. pneumoniae* and *E. coli* were generated using standard methods in molecular biology. Both genes were amplified using PCR from the respective genomic DNA. Forward and reverse primers used in amplification of *E.coli nikA* (5′-GCGG<u>CCATGG</u>CTGCACCAGATGAAATC-3′ and 5′-GCGC<u>CTCGAG</u>TTAAGGTTTCACCGGTT-3′) contained *Nco*I and *Xho*I restriction sites for ligation into pET-28b. In the case of *nikA* from *K. pneumoniae*, the sequence encoding an N-terminal Strep-II-tag (Trp-Ser-His-Pro-Gln-Phe-Glu-Lys) and tobacco etch virus (TEV) protease cleavage site (Glu-Asn-Leu-Tyr-Phe-Gln) was included in the forward primer(5′-GCGC<u>CCATGGG</u>CTGGAGCCATCCGCAGTTTGAAAAAGAAAACCTGTATTTTCAGAGC GCGCCGTACCAGCTCAC-3′ and 5′- GCGC<u>CTCGAG</u>TTATGGCGTTACCGGGG-3′)[63]. To generate a C-terminally FLAG-tagged form of *K. pneumoniae nikA*, the gene was amplified from genomic DNA containing *Bam*HI and *Xho*I restriction sites (5′-CGCG<u>GGATCC</u> ATGTCTATTATCCGCCT-3′ and 5′GCGC <u>CCATGG</u>ATGGGCTGGAGCCATCC-3′). The nucleotide sequence encoding the FLAG tag was included in the reverse primer. The PCR product was inserted into pGDP-1 in-frame with a *bla* promoter for constitutive expression. The final construct was transformed into *K. pneumoniae* ATCC 33495.

### Galleria mellonella infection model

Overnight cultures of *K. pneumoniae* ATCC 33495 or *S. aureus* USA300 were grown in LB, washed twice in PBS, and adjusted to $OD_{600}$ values of 1.0 and 2.0, respectively (~$2 \times 10^8$ and $1 \times 10^8$ CFU, respectively). The cell suspensions were mixed at an equal volume of 300 μg/mL Zn-AMA before injection. The larvae were each injected with 10 μL (~200 μg) of bacterial suspension with and without Zn-AMA or PBS. Ten larvae were used per group, incubated at 30 °C, and monitored on a 24-hour cycle. Fresh *G. mellonella* larvae were obtained from The Dragon Lair (www.thedragonlair.ca).

### Cytotoxicity assays

HEK293 cells (ATCC CRL-1573; generation 15) were seeded at 7500 cells/well in 384-well tissue culture-treated white plates in 50 μL

Dulbecco Modified Eagle Medium (DMEM) supplemented with 10% fetal bovine serum (FBS), 2 mM L-glutamine, 100 μg/mL streptomycin, and 100 μg/mL penicillin. Cells were incubated for 18 h at 37 °C under 5% $CO_2$. After 18 hours, a total volume of 500 nL of AMA, Zn-AMA, Co-AMA, or Ni-AMA (2-fold dilutions) and water was added to the cells using a Labcyte Echo acoustic dispenser (compound; Beckman Coulter) and a combi nL (water; ThermoFisher). Cells were incubated for an additional 48 h, after which cell viability was assessed using Promega Cell Titer Glo 2.0 reagent (Fisher Scientific). Cell Titer Glo (50 μL) was added directly to the media using the combi nL; the plates were shaken for 2 minutes and then incubated for 10 minutes at room temperature. The luminescence was read on a Neo2 plate reader (Biotek) using luminescence fiber. Controls were untreated cells and cells treated with water only.

### Evaluation of urease-dependent pH changes in whole cells and cell-free extracts

*K. pneumoniae* whole-cell urease activity was determined with artificial urine containing 0.2% (w/v) casamino acids and 0.01% (w/v) phenol red in 96-well plates. For the growth of *S. aureus* in artificial urine, the medium was supplemented with 10% LB. After 20–24 hours of growth, bacteria were removed using 0.45 μm filter plates, and the final $A_{550}$ of the medium was measured using a plate reader.

To measure urease activity in cell-free extracts, overnight cultures (5 mL) of *K. pneumoniae* were grown in LB at 37 °C and collected by centrifugation ($5000 \times g$, 10 min) resuspended in 25 mM HEPES-NaOH, 150 mM NaCl, pH 7.5. The cells were subsequently lysed by sonication, and cell debris was removed with centrifugation ($20,000 \times g$, 10 min). The cell-free extract was used in dose-response assays containing twofold dilutions of AMA and Zn-AMA (1 mM–3.9 μM), or fluorfamide (125 μM–61 nM) in flat bottom 96-well plates. The assays were initiated with urea (40 mM) and incubated at 37 °C for 10 min. Liberated $NH_3$ was quantified using the indophenol-hypocholorite reaction as previously described[64]. The total amount of protein was quantified by Bradford assay using bovine serum albumin as standard. Each assay was performed in duplicate wells.

### Polystyrene microtiter biofilm assay

Biofilm assays to assess the effect of Zn-AMA were adapted from a previously described method[65]. Overnight cultures of *K. pneumoniae* or *S. aureus* USA300 were diluted in sterile saline (NaCl, 0.9% w/v) to an $OD_{600}$ of 0.1 and diluted 1/200 in either M9-Cas, or artificial urine. Two-fold dilutions of Zn-AMA were added to the cells in a 96-well round-bottom microtiter plate, and the cells were grown at 37 °C for 18 hours in a static incubator. Following incubation, the microtiter plates were agitated on an orbital shaker (300 rpm, 2 min), the planktonic cells were transferred to a fresh plate, and their $OD_{600}$ was quantified. The biofilms were washed three times with deionized water, air-dried, and stained with 0.1% crystal violet for 15 min. Excess crystal violet was removed, the plates were washed with deionized water three times, and they were air-dried for 1 hour. The crystal violet-stained biofilms were solubilized in 33% acetic acid (100 μL) for 5 min, and the absorbance (600 nm) was measured. Experiments were performed in four replicates, and absorbance measurements were performed in a BioTek Synergy H1 plate reader.

### Whole-cell [NiFe] hydrogenase assay

The whole-cell hydrogenase activity in *K. pneumoniae* was based on a modified method described by Lacasse et al.[41] Fresh colonies of *K. pneumoniae* were resuspended in sterile saline of 0.85% (w/v) until an $OD_{600}$ of 0.1 was attained. The cell suspension was diluted 1/100 in LB supplemented with 0.8% glucose, 10 mM $MgCl_2$, and 10 mM potassium phosphate, pH 7.2. The cells were grown in 150 μL volumes in flat-bottom 96-well plates in an anaerobic glove box (85% $N_2$, 10% $CO_2$, 5% $H_2$) and incubated at 37 °C. Dose-response analyses of AMA and AMA

metal complexes were performed using twofold dilutions of each compound (200–0.78 μM). Following an 18-hour incubation, whole-cell [NiFe] hydrogenase activity was monitored by adding benzyl viologen (1 mg/mL) to each well, followed by continuous monitoring of the change in absorbance at 630 nm using a BioTek Synergy H1 microplate reader in an anaerobic glove box.

## [NiFe] hydrogenase zymography

As described for whole-cell hydrogenase assays, *E. coli* BW25113 cultures (2 mL) in eight-well cell culture plates were grown under anaerobic conditions. The cells were harvested by centrifugation, resuspended in PBS, and lysed by sonication. Cell debris was removed by centrifugation ($21,000 \times g$, 5 min), and Triton X-100 was added to a final concentration of 1%. Total protein was separated using a 12% native PAGE gel containing 0.1% Triton, and [NiFe] hydrogenase gel bands were developed using benzyl viologen (0.5 mg/mL) and tetrazolium chloride (1 mg/mL) in 50 mM MOPS pH 7 in a resealable bag that was flushed and filled with 100% hydrogen. Bands were detectable within 10 mins of incubation[42].

## Cellular metal analysis

The cellular $Zn^{2+}$ and $Ni^{2+}$ content of *K. pneumoniae* was carried out according to the method described by Maunders et al.[66] with some modifications. An overnight culture of *K. pneumoniae* was diluted 1/200 in LB medium (5 mL) and was either untreated or treated with 0.2 mM AMA or Zn-AMA. The cells were grown for 16 h at 37 °C, harvested by centrifugation, and washed twice with PBS containing 5 mM EDTA and twice with ultra-pure water. The cell pellet was subsequently digested in 0.25 mL 65% $HNO_3$ at 80 °C for 30 minutes in borosilicate glass tubes using a water bath. The samples were diluted to 2% $HNO_3$ and filtered through a 0.45 μm syringe filter. $^{60}Ni$ and $^{66}Zn$ were quantified using a Thermo iCAP Q ICP-MS.

## Cellular thermal shift assay

Overnight cultures of *K. pneumoniae* constitutively expression NikA-FLAG were diluted 1/20 in fresh LB (20 mL) supplemented with 50 μg/mL kanamycin and grown to an $OD_{600}$ of 0.85. The cells were subsequently chilled on ice, collected by centrifugation ($5000 \times g$, 10 min, 4 °C), resuspended in PBS (3 mL), and split into 1 mL aliquots to which water, AMA (250 μM), or Ni-AMA (250 μM) was added. The cells were then incubated at 37 °C for 20 min before being collected and washed twice in PBS by centrifugation. The final washed cell pellet was resuspended in PBS (100 μL) and aliquoted (20 μL) into PCR tubes. The cells were subjected to a temperature gradient using a thermocycler for 3 min, followed by a 3 min incubation at 25 °C. Total cellular protein was released by lysis with PBS containing 50 μg/mL lysozyme, 150 U/mL benzonase, 0.8% (v/v) NP-40, 1× EDTA-free protease inhibitor cocktail tablet, and 1 mM $MgCl_2$ (20 min), followed by three freeze-thaw cycles in liquid nitrogen and a 30 °C heating block. Cell debris was removed by filtration through a 0.45 μm 96-well filter plate, and 10 μL 4× SDS-PAGE loading dye was added. The samples were analyzed by SDS-PAGE and transferred onto a PVDF membrane for immunoblotting. Membranes were blocked with 5% non-fat skim milk, probed for FLAG-tagged NikA with mouse-derived anti-DYDDDDK IG2b conjugated to HRP (1:5000), and were detected with chemiluminescence with a ChemiDoc MP imaging system. The membrane was treated with hydrogen peroxide to inactivate HRP, blocked, reprobed with rabbit anti-OmpA IgG (1:10,000) followed by HRP-conjugated mouse anti-rabbit IgG (1:20,000) and quantified with chemiluminescence.

## Protein production and purification

*K. pneumoniae nikA*, lacking its N-terminal signal peptide (residues 20-518), was cloned into pET-28b with an N-terminal Strep-II tag. The plasmid was transformed into *E. coli* BL21 (DE3) for protein production, and an overnight culture was used to inoculate 1 L of LB supplemented with 50 μg/mL kanamycin, which was grown to an $OD_{600}$ of 0.6 at 37 °C. The culture was subsequently chilled to 18 °C, and isopropyl-β-thiogalactoside was added (0.5 mM) to induce expression. The culture was grown for an additional 18 h, then collected by centrifugation, resuspended in lysis buffer (100 mM Tris-HCl, 150 mM NaCl, pH 8.0), and lysed by sonication. The crude lysate was cleared by centrifugation ($30,000 \times g$, 15 min, 4 °C), and the total cellular protein was applied to a 1 mL StrepTrap XT column with a syringe. The column was washed with 10 column volumes (CV) of lysis buffer until no protein was detected in the flow through. Strep-II NikA was eluted from the column using five CV lysis buffer containing 50 mM biotin and was >90% pure (Fig. S16). Protein fractions were pooled, concentrated, and buffer exchanged into 25 mM Tris-HCl, 150 mM NaCl, pH 7.5 using a PD-10 desalting column. Using the calculated molar extinction coefficient, *K. pneumoniae* NikA was quantified using UV absorbance at 280 nm.

NikA from *E. coli* was produced and purified based on the method of Cherrier et al.[45]. Briefly, *E. coli nikA* was cloned into pET-28b, lacking its N-terminal signal sequence (22–524), and was kept tagless. *E. coli* NikA was overproduced as *K. pneumoniae* NikA and lysed in 40 mM Tris-HCl, 150 mM NaCl, pH 7.4. NikA was precipitated from cell lysates using 80% (w/v) ammonium sulfate, following a 40% (w/v) cut and dialyzed in 25 mM Tris-HCl, pH 8.0. NikA was further purified using anion exchange chromatography with a RESOUCE Q column and eluted with a gradient of NaCl. Peak fractions were pooled, and the >90% pure fractions were pooled (Fig. S16). The *E. coli* NikA was quantified with UV absorbance.

## In vitro thermal shift assay

*K. pneumoniae* NikA (1 μM) was incubated with AMA, Ni-AMA, Zn-AMA, Co-AMA, $NiCl_2$, $CoCl_2$, or $ZnCl_2$ (100 μM) in 25 mM Tris-HCl, 150 mM NaCl, pH 7.5 containing 2 X SYPRO Orange Dye. Thermal denaturation was monitored using an RT-PCR system (BioRad) between 25 °C and 95 °C at 0.5 °C increments. The melting temperatures of each sample were determined by identifying the inflection point of the derivative data of the melt curve.

## Isothermal titration calorimetry

Affinity and thermodynamic analyses of *K. pneumoniae* NikA were performed on a Malvern MicroCal PEAQ microcalorimeter. Titrations were performed in 25 mM Tris-HCl, 150 mM NaCl, 0.1% (*v/v*) Triton X-100, pH 7.5. It was found that the inclusion of detergent significantly deterred protein precipitation caused by the titration of $Zn^{2+}$ complexes. Compounds were prepared in the same buffer. NikA (15 μM) was loaded into the sample cell, and the various metal complexes (250 μM) were loaded into the syringe as ligands. In the case of Zn-toxin and AMB, 350 μM of the ligand was used. Titrations were conducted in duplicate at 37 °C with constant stirring at 750 rpm with an initial injection of 0.4 μL followed by 18, 3 μL injections. Control titrations to determine the heat of compound dilution into buffer were performed under the same conditions and subtracted from the experimental titration. Data were fit to a one-site model to determine the thermodynamic parameters of binding.

To determine the $Ni^{2+}$ affinity of AMA, competitive titrations were carried out with AMA, $Zn^{2+}$, and $Ni^{2+}$ using 25 mM Tris-HCl, 150 mM NaCl, pH 7.5. The sample cell was loaded with a $ZnCl_2$ and AMA at a 2:1 ratio using 200 μM and 100 μM, respectively. $NiCl_2$ (1 mM) was loaded into the syringe for titration into the sample cell. Titrations were performed in duplicate as described above. Data were fit to a one-site model competitive binding model.

## Crystallization, data collection, and structure determination

*E. coli* NikA was concentrated at 10 mg/mL, to which Ni-AMA was added at a final concentration of 400 μM (1:2 protein to ligand ratio). The

sample was then immediately used in sparse matrix crystallization screens with the hanging-drop vapor diffusion method with 2 μL drops and protein to reservoir ratio of 1:1. Needle-like crystals were identified in condition 9 (0.17 ammonium acetate, 0.085 sodium citrate pH 5.6, 25.5% PEG 4000, 15% glycerol) of the Hampton Research Crystal Screen Cryo suite after one week at 21 °C. The crystals were further optimized, and the best that grew as long rods in 0.17 M ammonium acetate, 0.1 M sodium citrate pH 5.8, 23% PEG 4000, and 17% glycerol were vitrified in liquid nitrogen. Native datasets were collected at the CMCF-BM (08IB1) beamline at the Canadian Light Source, Saskatoon, SK, Canada. The X-ray data were processed using autoPROC[67], XDS[68], and CCP4[69]. The structure of NikA was determined by molecular replacement using the NikA:Ni-(His)$_2$ complex (PDB ID 4I8C) as the search model. Model building and refinement were done in Coot[70] and Phenix[71] with translation/libration/screw (TLS) groups determined automatically using the TLSMD webserver[72]. Ramachandran statistics were calculated using Phenix using Molprobity, which gave 97% favored and 0.2% outliers. Ligand restraints for AMA generated using the Grade-WebServer (http://grade.globalphasing.org). Data are listed in supplementary table S1. The coordinates and structure factors have been deposited (PDB ID 8SPM) in the Protein Data Bank, Research Collaboratory for Structural Bioinformatics, Rutgers University, New Brunswick, NY. Molecular graphics and analyses were performed using Pymol.

### Reporting summary

Further information on research design is available in the Nature Portfolio Reporting Summary linked to this article.

## Data availability

The coordinates and structure factors of NikA:Ni-AMA generated in this study are available at the PDB under the accession number 8SPM. The previously published coordinates for NikA apoprotein, NikA:Fe-EDTA, NikA-Ni(L-His)$_2$, and NikA:Ni-BTC are available at the PDB under the accession numbers, 1UIU, 1ZLQ, 4I8C, and 3DP8, respectively. Source data are provided with this paper.

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

## Acknowledgements

We thank John C. Whitney for providing crystallography resources, Caitlyn M. Rotondo for technical support in cloning *nik*A, Victoria E. Coles for helpful discussions, and Susan McCusker and Tracey Campbell for their technical support with the cytotoxicity assays. We would also like to thank the ANALEST facility at the University of Toronto for providing access to their ICP-MS system. This research was funded by a Canadian Institutes of Health Research grant (FRN-148463), a Canadian Institutes of Health Research Fellowship award (to David Sychantha), and a Natural Sciences and Engineering Research Council of Canada (NSERC) grant RGPIN-2018-04968 (to Gerd Prehna).

## Author contributions

D.S. conceived, performed, and analyzed data for all experiments. X.C. Conducted *G. mellonella* infection models. K.K. synthesized AMA analogs. G.P. performed data collection and analysis of X-ray data. G.D.W. conceived the project, analyzed data, and directed the work of D.S. G.D.W. and D.S. prepared the manuscript with support from the other authors.

## Competing interests

The authors declare no competing interests.
