## [Peer Review File · Nature Communications]

Targeting bacterial nickel transport with aspergillomarasmine A suppresses virulence-associated Ni-dependent enzymesReviewer #1 (Remarks to the Author):

The authors demonstrated that aspergillomarasmine A (AMA), a metallophore that binds to Zn, Ni, and Co dications, can sequester Ni away from bacterial cells and thus reduce the catalytic activities of urease and [NiFe]-hydrogenase, two Ni-dependent enzymes. This aspect of the work is not innovative since similar findings have been demonstrated using other Ni chelators such as EDTA and DMG.

The novel aspect of this work is that the inhibition was stated to be greater when using Zn-AMA or Co-AMA complexes, however that claim (supported by Fig. 2B/E) is inconsistent with Fig. 2C, S2, or S4. No explanation for the inconsistency is provided.

The authors suggest the Zn and Co compounds inhibit the uptake of Ni into the cells, but no direct analysis of cellular Ni contents or Ni transport rates were included.

The metal complexes of AMA were shown to bind to NikA, a periplasmic protein that assists with Ni uptake, in a manner much like the authentic NikA ligand, Ni-(L-His)₂, as shown by x-ray crystallographic structure determination. Altered versions of AMA were identified that, when metalated, bound more tightly to NikA than Zn-AMA. The cellular fate of the NikA-bound analogs was not examined.

The authors suggest that molecules such as these could be used as antimicrobial compounds by acting in a novel manner, i.e. by binding to NikA and preventing the uptake of Ni. This is an intriguing hypothesis and worthy of further study. The described experiments generally are performed well and clearly described but are often indirect, leading to overstatements of the significance of some findings.

Additional comments and minor concerns are listed below in the order of their presentation in the text:

Fig. 1: Is it known that Ni-(L-His)₂ is needed for transporting Ni through the OM, which contains porins, or is this complex only required for IM transport? Do the colors of the components have any meaning?

Fig. 2:

(A) Is it established that the metal is 5-coordinate, as shown, for each of the complexes? Line 145 says these have octahedral coordination. Could a water molecule serve as a 6th ligand? If so, this should be shown. The term "subunits" in the legend could be confusing and should be replaced. An uppercase L should be used (twice) with Asp.

(B) Was cell growth affected by addition of the various compounds? What phenol red absorbance value corresponds to 100 on the scale? The authors understand these 24 h timepoints of phenol red absorbance are indirect measures of urease activity as shown by their use of the phrase "qualitatively represent"; the absorbances are unlikely to directly correlate with urease specific activity which was not measured. (D) Same comments as for B.

(E) What benzyl viologen absorbance value corresponds to 100% relative hydrogenase activity? Can this be correlated to enzyme specific activity? Did the LB-glucose contain Ni in this experiment, analogous to what is specified for the medium in panel F?

Fig. S1: Define DP, in the y-axis. The data show the *K_d* (note that thermodynamic constants should be shown in italics) of AMA for Zn is 200 nM and that for Ni is 80 nM; however, the text (line 119) states that the value for Ni is 80 pM and 2-fold higher than for Zn. These inconsistencies are concerning.

Fig. S2: The 24 h change in phenol red absorbance at 550 nm is unlikely to represent a true measure of the relative urease activity in the buffered medium. The axis labels should be corrected to indicate measurement of the change in absorbance values, or the authentic urease specific activities should be measured for the two strains. Fix "a change in resulting from a pH change". Also, the data in panel A do not appear to agree with the data in Fig. 2B; here AMA is most effective whereas Zn-AMA then Co-AMA are more effective than AMA in Fig. 2B. How is this

explained?

Fig. S3: Again, the measurements do not truly depict the relative urease activities. The legend incorrectly indicates "AMA and its metal complexes" and again says "a change in resulting from a pH change".

Fig. S5: A positive control with AHA would have been useful to show that urease inhibition can be detected with this assay. Again, however, the results do not really represent relative urease activity but only indicate the change in phenol red absorbance in this buffered solution. The legend indicates the "rate of change in the absorbance" was measured, but the time points used for this measurement were not specified.

Line 170: The authors showed that Zn-AMA acted upstream of urease itself, but at this point in the text they did not demonstrate it acted "upstream of Ni²⁺ accessory proteins".

Fig. 3:

(A) Again, urease activity is not directly measured, rather the data show a urease-induced pH change for the 24 h single time point. The legend should use an uppercase L for L-His (four times). (E/F) Define the y-axis (-dRFU/dt). This axis seems less intuitive than directly plotting relative fluorescence units, the preference of this reviewer, but perhaps it is more sensitive for identifying the transition temperatures. Including Ni-(His)₂ in this set of studies would be informative. The mention of "RT-PCR system" may be confusing to readers and is not needed since no RT-PCR is carried out here.

Fig. S7: Again, consider showing the more intuitive RFU vs temperature plots and removing mention of RT-PCR.

Fig. 4: Do any of the metal complexes have water as a sixth ligand in solution? Are all these 5-coordinate structures established? Does "slightly altered geometry" (line 215) include differences in coordination number? Use uppercase in the legend for L and for Triton.

Fig. 5: Delete "from" in the first sentence of the legend. Why is the His ligand included in the omit map of panel B? Perhaps it is disordered in the Ni-free state? Why is the H416 ligand not shown in panel C? The unbound Ni-AMA has a carboxylate ligand from the adjacent complex in its crystal structure, raising the possibility of solvent coordination in solution; is the structure of that molecule known and should water be shown?

Fig. 6: Since the figure title mentions "known ligand-bound structures", comparison also could be made to the structure of NikA with Fe-EDTA (PDB 1zlq); does that structure coincide better with the open or closed form? Panel C does not indicate anything about whether H416 is bound to these complexes (it is not for BTC or the Fe-EDTA structure), but this aspect should be included. The legend includes the lab jargon "apo NikA" which should be corrected to "NikA apoprotein". Use uppercase for L in L-His (three times and shift the hyphen in the first case).

Fig. 7: NikA and a period were missing from the figure title. Reasonable chemical structures are shown for the compounds, but the true coordination geometries are unknown and water ligands may be present in some cases; this potential uncertainty should be indicated.

Fig. S8: In the figure title, clarify that the titration was versus NikA (not Zn titration with the AMA analogs). Fix "were performed were performed". Use uppercase for Triton.

Line 324: The mention that "Ni-binding proteins belong to two of three structural classes of the periplasmic binding protein superfamily" is confusing and seems to serve no purpose since the classes are not further described. Perhaps delete or expand.

Line 336: The fate of Zn-AMA could be clarified by quantifying its cytoplasmic levels using MS.

Other minor comments:

48, add comma after ligand; 82-83, change to "functions ... play"; 149, replace "chemical

interaction with AHA" with "physiological interaction with AHA" or other wording that avoids the suggestion of direct binding between AMA metal complexes and AHA; 161, change "media" to "medium" here and about 10 later instances; 185, reword "Ni-AMA transports Ni²⁺ into the cell" to "cells transport Ni²⁺ from Ni-AMA" since Ni-AMA cannot do action; 224 and 251, the PDB file is 4i8c and not 418c; 228-229, "subunit" (used 3 times) is a poor word choice here and can simply be deleted; 285, replace "subunit"; 331, the basis for stating AMA is pentadentate is unclear especially considering that water may be bound; 379, change "in a frame" to "in frame"; 420, replace "for" with "to an" or "until reaching an"; 444, change "isothio" to "isopropylthio"; 457, change "precepted" to "precipitated"; 498, change "Ni-(His₂)" to "Ni-His₂"; 500, define TLS; 503, the line about supplementary figure 1 is incorrect.

Reviewer #2 (Remarks to the Author):

The manuscript by Sychantha et al. is a well-written exploration of a critical issue: antimicrobial resistance. There is an urgent need for the development of novel strategies to target antibiotic-resistant infections, and this paper offers some preliminary findings towards a unique approach. Specifically, it focuses on the use of the metallophore aspergillomarasmine A (AMA) and Zn-AMA as a potential treatment for bacterial infections. Notably, the group had previously demonstrated the effectiveness of this compound as a beta-lactamase inhibitor. In this study, they extend its utility, now exploring its potential as a chemotherapeutic treatment.

The compound Zn-AMA has been identified as a promising nika inhibitor, but this manuscript has not sufficiently demonstrated its potential as a chemotherapeutic agent for treating bacterial infections. Key aspects are lacking in the study. First, an investigation into the compound's cytotoxicity was absent, leaving unconfirmed whether or not any existed. Second, the limited structure-activity relationship (SAR) studies did not enhance the compound's efficacy, suggesting the compound may have limited medicinal chemistry value. Finally, the paper did not include any attempt to showcase the application of this compound in an animal model, a necessary step to substantiate its intended use as a chemotherapeutic treatment of bacterial infections.

Therefore, I'm unable to recommend this manuscript for publication in Nat Com at this stage.

Other comments and suggestions:

AMA as a urease inhibitor:

AMA demonstrated significant inhibition of urease activity as noted on line 134, which would have been an intriguing avenue for further exploration. Given that AMA and Zn-AMA appear to operate on separate pathways, the potential synergistic relationship between these two compounds could have been an interesting area of study. This has already been explored in the case of AHA and Zn-AMA, so it would have been instructive to see a similar examination conducted for AMA and Zn-AMA.

Mode of action for Zn-AMA:

While Zn-AMA clearly binds to nika, lending credibility to claims of it being the intended target, the decrease in biological activity when Zn-AMA's binding affinity for nika is enhanced raises questions. It suggests the possibility that Zn-AMA may have an alternative or additional target, which may be worth exploring in further research.

Nika as a therapeutic target:

The manuscript falls short in substantiating why nika should be considered a compelling therapeutic target. Questions about the effects of nika knockout remain unanswered. It's unclear whether nika mutants can establish an infection or the potential impact this could have in vivo. While the importance of Ni-dependent enzymes is mentioned, there's a noticeable lack of detailed information about nika itself, which is crucial given that it's proposed as your target. Thus, more specific insights about nika should be incorporated.

Reviewer #3 (Remarks to the Author):

In this manuscript, Sychantha et al. report that a fungal metallophore (aspergillomarasmine A, AMA) has high affinity for divalent cations, including Zn^{2+} , Ni^{2+} , and Co^{2+} . The use of AMA alone or in combination with the aforementioned metallic cations resulted in inhibition of two well-characterized nickel-containing enzymes, urease and [NiFe] hydrogenases in various organisms. Urease activity was inhibited by AMA in *K. pneumoniae* and *S. aureus* (as shown by measuring enzyme activity), resulting in lower levels of urease-generated struvite crystals in both pathogens. Likewise, (H_2 -producing) [NiFe] hydrogenase activity was found to be inhibited in *K. pneumoniae* and *E. coli*. Using a variety of biochemical approaches, the authors do demonstrate that Zn-AMA and Co-AMA bind to NikA, the periplasmic component of the NikABCDE nickel transporter, thereby competing with the natural nickelophore L-His. This is a significant finding. They further characterized the NikA-Ni-AMA binding by X-ray crystallography. Finally, they hypothesized that AMA could be used as a novel antibacterial agent against Ni-requiring pathogens, by inhibiting Ni-enzymes.

The reviewer agrees with the authors that nickel sequestration is a promising avenue to combat antibacterial-resistant pathogens. While this avenue has been already explored by several research groups, the novelty herein is the use of the fungal metallophore AMA. The study is interesting, and contains some significant work, the bulk of the methods used are adequate, and the conclusions are largely supported by results. However, the authors failed to provide any convincing results that would link the use of AMA to growth inhibition and/or virulence attenuation of the various pathogens they study here (e.g., *S. aureus*, *K. pneumoniae*). That addition would make the study complete.

The fact that AMA inhibits Ni-enzymes suggests it might have an effect on the growth of Ni-requiring pathogens. Certainly, growth conditions exist that rely on specific hydrogenases and on urease. Results on *K. aerogenes* reported herein are promising, however they are preliminary, and AMA needs to be tested on other pathogens used in this study (*S. aureus*, *S. Typhimurium*, *K. pneumoniae*), including MDR strains, if possible.

A main concern is about the effect of AMA on [NiFe] hydrogenases; I believe this avenue was not properly studied (see below). What type of hydrogenase is inhibited and certainly by how much is not clear. So, my overall impression of this work is mixed, and the study needs to be more thorough.

Major comments

There is lack of basic understanding of hydrogen metabolism, hydrogenases and even nickel homeostasis as a whole throughout the manuscript. Some background needs to be introduced here. Even though the term "hydrogenases" collectively refer to enzymes that can either produce or use H_2 , H_2 -uptake (oxidizing) hydrogenases and H_2 -producing hydrogenases are vastly different enzymes, both in structure and activity, and should be considered as two different types of enzymes, based on their respective contribution, e.g., H_2 -uptake hydrogenases, such as Hya (Hyn) or Hyb, are respiratory enzymes that generate electrons (to feed the ETC) and protons (to generate PMF and ATP), whereas H_2 -producing hydrogenases, such as Hyc and Hyf, are integral parts of larger complexes (FHL and FHL-2), whose main role is to dispose of reductant. The number of hydrogenases within a given organisms, and the type of reactions they perform, is of paramount importance for both growth and virulence in the case of pathogens. Yet, throughout the text, Sychantha et al. fail to specify which type and how many [NiFe] hydrogenases are present in the organisms they study, always referring to "[NiFe] hydrogenase" as if it was one type of enzymes, performing one type of reaction. This lack of precision/accuracy results in misleading statements, incomplete experimental designs, and partial conclusions, as well as a discussion that does not mention anything about hydrogenases!

1. abstract, line 35: "critical enzymes involved in colonization and persistence, including urease and [NiFe]-hydrogenase (singular)..."
2. Introduction, lines 80-81: "Similarly, [NiFe]-hydrogenases catalyze the oxidation of hydrogen (H₂) under anaerobic conditions to generate energy in nutrient-poor environments, as observed for *H. pylori* and *S. Typhimurium*". There are many inaccurate statements in this sentence: first, this suggests that all [NiFe] hydrogenases are H₂-uptake enzymes, which is not the case. Second, anaerobic conditions are not required for H₂-uptake [NiFe] hydrogenases to be active. Third, the two organisms cited as examples could not be more different, as *H. pylori* only possess one (H₂-uptake) hydrogenase, whereas *S. Typhimurium* contains four, including three H₂-uptake types and one H₂-evolving enzyme.
3. Introduction, line 85: "The enzymatic activities of urease and [NiFe]-hydrogenase depend...).
4. Results, lines 161-162: "Dose-response assays were performed on bacteria grown in rich media (LB-glucose) using strict anaerobic conditions under which [NiFe]-hydrogenases are produced. Which [NiFe]-hydrogenases are the authors referring to? Under strict anaerobic conditions, and in absence of added H₂, they are likely to monitor only the H₂-evolving hydrogenase, whose contribution to growth and pathogenicity has been shown in previous studies to be negligible.
5. Results, lines 166-167:" The distinct effect of these compounds on [NiFe]-hydrogenase activity was verified with zymographic analysis of *E. coli* cell lysates". Which [NiFe]-hydrogenases are the authors referring to? The M&M section specific to the zymograms study mention the use of H₂ and BV. In this condition, they probably observed H₂-oxidizing activity; if so, corresponding enzymes should be described.
6. Methods, lines 397-406, section on "Whole-cell [NiFe] hydrogenase assay". References are missing, and probably for a good reason; I have not seen this protocol before, and the aim of this assay is not clear. The experimental design chosen by the authors (anaerobic growth with no H₂ provided) suggests that they expect to monitor H₂-producing hydrogenases (Hyc or Hyf), however these usually have negligible activity, and they have been shown to be the least important both in terms of growth and virulence of pathogenic bacteria (see previous published work on *S. Typhimurium* hydrogenases). In addition, if the goal was to monitor H₂ production, then reduced BV should be used (as electron donor), not oxidized BV---an electron acceptor is not needed for H₂ production assays. In summary, it is not clear to me what they are trying to achieve here, but these results are incomplete at best.
7. In the discussion section, the authors do not discuss the potential and limitations of using AMA alone or with metals to inhibit hydrogenase activity/activities. In fact, hydrogenases are not mentioned at all in the discussion section.

Minor comments:

A number of sentences are vague; this includes the title of the manuscript, paragraph headings, statements, figure legends etc; sometimes there is no transition between paragraphs or conclusions. Also, the authors describe nickel homeostasis in various organisms (*K. pneumoniae*, *E. coli*, *S. aureus*), however often it is not clear which one they refer to when reporting results or describing previously described mechanisms. A few examples:

1. The title "Targeting bacterial nickel transport inhibits virulence-associated Ni-dependent enzymes" reads like a review title, and does not let readers expect anything novel, since (i) inhibiting nickel transport is expected and has already been shown to result in Ni-enzyme inhibition; and (ii) several Ni-enzymes (hydrogenases and urease) have already been shown to be required for growth and virulence in several pathogens. The novelty here is the use of the fungal metallophore aspergillomarasmine A to inhibit Ni-enzymes and Ni-requiring pathogens, so this should be included in the title.
2. Abstract, lines 38-39 " AMA may block Ni²⁺ uptake and attenuate Ni-dependent enzymes as a potential anti-virulence strategy" does not read well
3. Abstract, lines 51-52: " These findings...in the chemotherapeutic treatment of bacterial infections that depend upon Ni²⁺ as a micronutrient". Bacteria depend upon Ni as a micronutrient, bacterial infections do not.
4. Introduction, lines 86-91 "The ABC-type transporter (NikABCDE) imports the metal from the extracellular environment..." which organisms are the authors referring to in this statement? It comes right after describing the role of hydrogenase in *H. pylori* and is written as if NikABCD was the universal way of importing Ni²⁺ into the cytoplasm; however, some bacteria, including *H.*

- pylori, use other types of high affinity transporters (NixA, Niu). Please be more specific.
5. Introduction, line 59 . "linking the role of Ni-dependent enzymes to the colonization..., not in the colonization..."
 6. Introduction, line 61" Salmonella enterica serovar Typhimurium, or Salmonella enterica Typhimurium" when introduced for the first time.
 7. Introduction, lines 64-65. Please provide one or several references for the statement on calprotectin.
 8. Fig. 1. "The role of nickel in virulence". This Fig. title is vague. Which organism are they referring to? Gram negative obviously, since an outer membrane is depicted? E. coli? K. pneumoniae? Please be more specific.
 9. Fig. 3. "Bacterial nickel uptake is inhibited by competitive binding to the periplasmic Ni-binding protein NikA". Which organism are the authors referring to?
 10. Results, lines 120-121: "The K_d value for Ni²⁺ was estimated to be 80 pM, 2-fold higher than AMA's affinity for Zn²⁺ and is consistent with the Irving-Williams order of ligand strength for these metals (Fig. 2A and Fig. S1). First, the authors report a K_d value of 80 pM for Ni-AMA and 200 pM for Zn-AMA, which is 2.5-fold higher, not 2-fold higher. Of concern, both K_d values are in the pM range (text, and Fig. 2A) whereas they are reported in the nM range (ITC results, Fig. S1).
 11. Results, lines 143-144: "These results confirm our hypothesis and the ability of AMA to suppress urease activity while also suggesting a mechanism beyond Ni²⁺ sequestration". I do not understand this sentence. Please rephrase.
 12. Results, lines 148-157. The authors discuss two types of mechanisms by which Zn-AMA and AHA could inhibit urease activity. One involves "synergism", whereas the other would be "additive effect". In my opinion, synergism and additive effect mean the same.
 13. Results, lines 150-151 "Zn-AMA inhibited urease indirectly by interfering with urease maturation in K. pneumoniae by disrupting Ni²⁺ uptake" does not read well.
 14. Discussion, line 317: what is an "untargeted" mechanism?
 15. Methods, lines 387-395. Urease activity in cell-free extracts has to be standardized to protein concentration. Even when using the same method to break cells, one cannot expect the lysis to yield the same amount of protein, including urease; this is likely to have an effect on the final enzyme activity levels.
 16. Methods, line 440: "K. pneumoniae NikA...was cloned into pET28b..." A gene (nikA) (not a protein) is cloned into a plasmid.
 17. Methods, line 444 "β-galactoside" needs to be corrected
 18. Methods, line 455, same as above "NikA cloned into pET-28b..." A gene (nikA) is cloned, not a protein. Also, I believe a "was" is missing
 19. Fig. 3. "Bacterial nickel uptake is inhibited by competitive binding to the periplasmic Ni-binding protein NikA". Which organism are the authors referring to?
 20. Results presented in Fig S3 and Fig S5 are confusing, Fig. legends are vague and do not give an accurate description of the experiment. Fig S3 legend: "Dose-response analysis of ZnCl₂ on urease activity of K. pneumoniae" does not mean anything, please rephrase. Fig. S3 shows the effect of ZnCl₂ on urease activity, but Fig S3 legend states "Urease activity was determined in artificial urine supplemented with 2-fold dilutions of AMA and its metal complexes at 37 °C for 24 hours." Did the authors study the effect of ZnCl₂ only on urease activity in K. pneumoniae? The effect of Zn-AMA? The effect of Zn in presence of AMA and other metal complexes?
 21. Fig. 1 Legend: "Ni-dependent enzyme" should be plural ("enzymes").
 22. Fig. 9 should be Fig. S9.
 23. Fig. S9 Legend. "Relative urease activity was measured as a change in resulting from a pH change in the media" does not read well and needs to be rephrased. I believe "absorbance at 550 nm" is missing". Also, "It was detected with an increase in absorbance at 550 nm using the pH indicator phenol red" is confusing. Is there still an increase in absorbance when authors report 0% relative urease activity? The difference in absorbance at 550 nm between T0 and T24 is what matters.
 24. Fig S1 legend refers to (A) and (B), but there is no (A) and (B) shown on Fig S1.
 25. Fig S3 legend "Dose-response analysis of ZnCl₂ on urease activity of K. pneumoniae" does not mean anything, please rephrase.
 26. Fig. S5 Legend. FIG S5. "Dose-response analysis of AMA and Zn-AMA in cell-free extracts of K. pneumoniae" does not mean anything, please rephrase. And "K. pneumoniae" needs to be corrected.
 27. A gel showing purified KpNikA and EcNikA recombinant proteins should be shown in the

supplement.

Reviewer #1 (Remarks to the Author):

The authors demonstrated that aspergillomarasmine A (AMA), a metallophore that binds to Zn, Ni, and Co dications, can sequester Ni away from bacterial cells and thus reduce the catalytic activities of urease and [NiFe]-hydrogenase, two Ni-dependent enzymes. This aspect of the work is not innovative since similar findings have been demonstrated using other Ni chelators such as EDTA and DMG.

The novel aspect of this work is that the inhibition was stated to be greater when using Zn-AMA or Co-AMA complexes, however that claim (supported by Fig. 2B/E) is inconsistent with Fig. 2C, S2, or S4. No explanation for the inconsistency is provided.

We thank the reviewer for highlighting this potential source of confusion. The maximum culture density of *K. pneumoniae* is lower in artificial urine than in LB media. This underestimated the extent of urease activity in the cells after 24h. To improve the growth of *K. pneumoniae* and increase total urease activity over 24h, artificial urine was supplemented with casamino acids and yeast extract. The revised data, replacing the originals in Fig. 2B and Fig S2A, clearly show that AMA is less effective than Zn-AMA and Co-AMA.

Instead of using relative urease activity values, the revised data now compare the effects of AMA, Zn-AMA, Co-AMA, and Ni-AMA on urease-dependent pH change. Based on this, Zn-AMA shows dose-dependent suppression of culture alkalization. After 24, the pH did not increase above 7.2 at 6.25 μ M Zn-AMA, consistent with the suppression of struvite formation in Fig. 2C and Fig. S4. AMA did slightly reduce alkalization but not enough to inhibit struvite formation under the revised conditions.

The authors suggest the Zn and Co compounds inhibit the uptake of Ni into the cells, but no direct analysis of cellular Ni contents or Ni transport rates were included.

We acknowledge the reviewer's comment. We have addressed it by quantifying cellular Ni and Zn content in *K. pneumoniae* and *S. aureus* using ICP-MS. The data confirmed that Ni levels were reduced in AMA- and Zn-AMA-treated bacteria, shown in Fig. 2E. (Lines 160 - 170)

The metal complexes of AMA were shown to bind to NikA, a periplasmic protein that assists with Ni uptake, in a manner much like the authentic NikA ligand, Ni-(L-His)₂, as shown by x-ray crystallographic structure determination. Altered versions of AMA were identified that, when metalated, bound more tightly to NikA than Zn-AMA. The cellular fate of the NikA-bound analogs was not examined.

We appreciate the reviewer's interest in the cellular fate of Zn-AMA. We are also interested in this because a better understanding of the cellular fate would certainly be helpful for the optimization of Zn-AMA analogs. However, as the specific details of Ni(L-His)₂ uptake are known (i.e. the interactions and dynamics between NikABCDE), determining the cellular fate of Zn-AMA falls outside the scope of this work. In the current state of the manuscript, determining the cellular fate of Zn-AMA would not change the study's overall conclusions. This is discussed in the manuscript (Lines 393-403).

The authors suggest that molecules such as these could be used as antimicrobial compounds by acting in a novel manner, i.e. by binding to NikA and preventing the uptake of Ni. This is an intriguing hypothesis and worthy of further study. The described experiments generally are performed well and clearly described but are often indirect, leading to overstatements of the significance of some findings. Additional comments and minor concerns are listed below in the order of their presentation in the text:

Fig. 1: Is it known that Ni-(L-His)₂ is needed for transporting Ni through the OM, which contains

porins, or is this complex only required for IM transport? Do the colors of the components have any meaning?

Previous work in *E. coli* has shown that the extracellular uptake of radioactive Ni²⁺ depends on exogenous L-His, presumably carrying it through the outer membrane. This mechanism of Ni-(L-His)₂ uptake was shown to be TonB-independent and suggested it crosses the outer membrane through non-specific outer membrane porins. A dedicated transporter has not yet been identified. The following reference has been added DOI: 10.1039/c2mt20139a

The colors of the components denote different protein subunits but are not assigned to specific functional classes. This has been clarified in the figure legend.

Fig. 2:

(A) Is it established that the metal is 5-coordinate, as shown, for each of the complexes? Line 145 says these have octahedral coordination. Could a water molecule serve as a 6th ligand? If so, this should be shown. The term "subunits" in the legend could be confusing and should be replaced. An uppercase L should be used (twice) with Asp.

Whether Zn²⁺ or Co²⁺ form a 5-coordinate complex has not been experimentally validated. However, the 5-coordinate structure of the Ni-AMA complex suggests that the stereochemistry of AMA would restrain the formation of 6-coordinate complex. While a water molecule could serve as 6th ligand, we have no information about its binding strength and stability. This has been addressed in the text (Lines 235-240).

(B) Was cell growth affected by addition of the various compounds?

No, all cells grew to the same OD600 as the untreated control. Spot dilutions of cells grown at the highest concentration of each compound were performed after 24 h, showing that these cells were still viable. (Figure S4)

The authors understand these 24 h timepoints of phenol red absorbance are indirect measures of urease activity as shown by their use of the phrase "qualitatively represent"; the absorbances are unlikely to directly correlate with urease specific activity which was not measured.

(D) Same comments as for B.

Attempts to quantify NH₃ production in living bacteria were unsuccessful because of high background values. To address the issue of using arbitrary changes in absorbance values, we have reported the relative pH change in the media.

(E) What benzyl viologen absorbance value corresponds to 100% relative hydrogenase activity? Can this be correlated to enzyme specific activity?

We thank the reviewer for this comment. To address it, we now show the specific activity values, calibrated with reduced benzyl viologen standards and reported. The new data are reported in Fig. 2F

Did the LB-glucose contain Ni in this experiment, analogous to what is specified for the medium in panel F?

To clarify, Ni was not included in the LB glucose used in Figure 2F. Ni was only added to cells used in zymograms to ensure enough activity could be detected in gel.

Fig. S1: Define DP, in the y-axis. The data show the K_d (note that thermodynamic constants should be shown in italics) of AMA for Zn is 200 nM and that for Ni is 80 nM; however, the text (line 119) states that the value for Ni is 80 pM and 2-fold higher than for Zn. These inconsistencies are concerning.

We thank the reviewer for catching this. DP in the y-axis has been defined (differential power), and the K_D values in Fig S1 were typos and should both be pM. These have been corrected.

Fig. S2: The 24 h change in phenol red absorbance at 550 nm is unlikely to represent a true measure of the relative urease activity in the buffered medium. The axis labels should be corrected to indicate measurement of the change in absorbance values, or the authentic urease specific activities should be measured for the two strains. Fix "a change in resulting from a pH change

As indicated above, the relative y-axis values have been replaced with pH values.

Also, the data in panel A do not appear to agree with the data in Fig. 2B; here AMA is most effective whereas Zn-AMA then Co-AMA are more effective than AMA in Fig. 2B. How is this explained?

As indicated above, urease activity was underestimated due to the poor growth of *K. pneumoniae* cells in artificial urine, which we improved by adding casmino acids to the media.

However, AMA still inhibits urease activity in *S. aureus*. Indicating that the complexed and uncomplex forms of the compound act on Ni-uptake through distinct pathways. This statement was made in the text. (Line 143-144)

Fig. S3: Again, the measurements do not truly depict the relative urease activities. The legend incorrectly indicates "AMA and its metal complexes" and again says "a change in resulting from a pH change

As indicated above, pH values are given.

Fig. S5: A positive control with AHA would have been useful to show that urease inhibition can be detected with this assay. Again, however, the results do not really represent relative urease activity but only indicate the change in phenol red absorbance in this buffered solution. The legend indicates the "rate of change in the absorbance" was measured, but the time points used for this measurement were not specified.

We thank the reviewer for this suggestion. Cell-free extracts were prepared, and urease-catalyzed urea hydrolysis was monitored by quantifying NH_3 production with a phenol-hypochlorite reaction. The known urease inhibitor fluorofamide was used as a control inhibitor. The specific activity values are now reported in Fig S6.

Line 170: The authors showed that Zn-AMA acted upstream of urease itself, but at this point in the text, they did not demonstrate it acted "upstream of Ni^{2+} accessory proteins".

This statement has been removed for clarity.

Fig. 3:

(A) Again, urease activity is not directly measured, rather the data show a urease-induced pH change for the 24 h single time point. The legend should use an uppercase L for L-His (four times).

As indicated above, the pH change is now reported. Corrections to the text have been in the legend of figure 3.

(E/F) Define the y-axis ($-\text{dRFU}/\text{dt}$). This axis seems less intuitive than directly plotting relative fluorescence units, the preference of this reviewer, but perhaps it is more sensitive for identifying the

transition temperatures. Including Ni-(His)₂ in this set of studies would be informative. The mention of "RT-PCR system" may be confusing to readers and is not needed since no RT-PCR is carried out here.

Fig. S7: Again, consider showing the more intuitive RFU vs temperature plots and removing mention of RT-PCR.

The y-axes have been defined in the revised figure legends. The negative derivative of the fluorescence allows the visualization of melting temperature more clearly. The mention of RT-PCR was removed in the figure legend.

Fig. 4: Do any of the metal complexes have water as a sixth ligand in solution? Are all these 5-coordinate structures established? Does "slightly altered geometry" (line 215) include differences in coordination number? Use uppercase in the legend for L and for Triton.

As indicated above, these structures have not been validated and are predictions. This is clarified in the manuscript (Lines 676-677). Slightly altered geometry refers to the pentadentate coordination geometry. More specifically, possible distortions in the metal-ligand distances. Clarification was included in the text (Lines 235-241). The typos were corrected.

Fig. 5: Delete "from" in the first sentence of the legend. Why is the His ligand included in the omit map of panel B? Perhaps it is disordered in the Ni-free state? Why is the H416 ligand not shown in panel C? The unbound Ni-AMA has a carboxylate ligand from the adjacent complex in its crystal structure, raising the possibility of solvent coordination in solution; is the structure of that molecule known and should water be shown?

His416 was excluded from the omit map because its occupancy was expected to be 1.0, but the reviewer's point about possible disorder was taken. An omit map including His416 was generated and included in the revised figure (Fig. 5B) The electron density for His416 in the mFo-DFc omit map is well defined.

For clarity, His416 was not included in panel C because its purpose was to highlight conformational changes. The His416 ligand is shown directly beneath panel C in panel D in a similar orientation, which shows the involvement of His416.

While the crystal structure of the unbound Ni-AMA has an adjacent complex filling the 6th site. It is an interaction induced by crystallization. While we agree that water could interact with the metal ion, as it would with any other functional group in the molecule, we do not know the stability of the interaction and whether rapid solvent exchange occurs. For this reason, we have chosen not to depict the structure of AMA metal complexes with bound water.

Fig. 6: Since the figure title mentions "known ligand-bound structures", comparison also could be made to the structure of NikA with Fe-EDTA (PDB 1z1q); does that structure coincide better with the open or closed form? Panel C does not indicate anything about whether H416 is bound to these complexes (it is not for BTC or the Fe-EDTA structure), but this aspect should be included. The legend includes the lab jargon "apo NikA" which should be corrected to "NikA apoprotein". Use uppercase for L in L-His (three times and shift the hyphen in the first case).

Thank you for the suggestion. To clarify, His416 does serve as a Ni ligand in the BTC-bound structure of NikA, but not in the EDTA-bound structure. These differences are now highlighted and compared in the manuscript. (Fig. 6B and C, and lines 285 – 299)

Fig. 7: NikA and a period were missing from the figure title. Reasonable chemical structures are shown for the compounds, but the true coordination geometries are unknown and water ligands may be present in some cases; this potential uncertainty should be indicated.

We agree with the reviewer that the structure of those complexes is not certain. For this reason, Figure 7 is converted to Table 1 in the revised manuscript. The uncomplexed structures are now shown to avoid misrepresenting the geometry of the complexes.

Fig. S8: In the figure title, clarify that the titration was versus NikA (not Zn titration with the AMA analogs). Fix "were performed were performed". Use uppercase for Triton.

This has been corrected.

Line 324: The mention that "Ni-binding proteins belong to two of three structural classes of the periplasmic binding protein superfamily" is confusing and seems to serve no purpose since the classes are not further described. Perhaps delete or expand.

This discussion point has been removed.

Line 336: The fate of Zn-AMA could be clarified by quantifying its cytoplasmic levels using MS.

This was comment was addressed above.

Other minor comments:

48, add comma after ligand; 82-83, change to "functions ... play"; 149, replace "chemical interaction with AHA" with "physiological interaction with AHA" or other wording that avoids the suggestion of direct binding between AMA metal complexes and AHA; 161, change "media" to "medium" here and about 10 later instances; 185, reword "Ni-AMA transports Ni²⁺ into the cell" to "cells transport Ni²⁺ from Ni-AMA" since Ni-AMA cannot do action; 224 and 251, the PDB file is 4i8c and not 418c; 228-229, "subunit" (used 3 times) is a poor word choice here and can simply be deleted; 285, replace "subunit"; 331, the basis for stating AMA is pentadentate is unclear especially considering that water may be bound; 379, change "in a frame" to "in frame"; 420, replace "for" with "to an" or "until reaching an"; 444, change "isothio" to "isopropylthio"; 457, change "precepted" to "precipitated"; 498, change "Ni-(His₂)" to "Ni-(His)₂"; 500, define TLS; 503, the line about supplementary figure 1 is incorrect.

We thank the reviewer for a thorough reading of the manuscript. After reviewing the minor comments and suggestions raised by the reviewer, we have accepted these critiques and incorporated all the appropriate changes.

Reviewer #2 (Remarks to the Author):

The manuscript by Sychantha et al. is a well-written exploration of a critical issue: antimicrobial resistance. There is an urgent need for the development of novel strategies to target antibiotic-resistant infections, and this paper offers some preliminary findings towards a unique approach. Specifically, it focuses on the use of the metallophore aspergillomarasmine A (AMA) and Zn-AMA as a potential treatment for bacterial infections. Notably, the group had previously demonstrated the effectiveness of this compound as a beta-lactamase inhibitor. In this study, they extend its utility, now exploring its potential as a chemotherapeutic treatment.

The compound Zn-AMA has been identified as a promising *nikA* inhibitor, but this manuscript has not sufficiently demonstrated its potential as a chemotherapeutic agent for treating bacterial infections. Key aspects are lacking in the study. First, an investigation into the compound's cytotoxicity was absent, leaving unconfirmed whether or not any existed. Second, the limited structure-activity relationship (SAR) studies did not enhance the compound's efficacy, suggesting the compound may have limited medicinal chemistry value. Finally, the paper did not include any attempt to showcase the application of this compound in an animal model, a necessary step to substantiate its intended use as a chemotherapeutic treatment of bacterial infections.

We thank the reviewer for their constructive comments. We have tested the cytotoxicity of AMA and its complexes with Zn, Ni, and Co *in vitro* against HEK293 cells. No significant toxicity was observed for AMA, Zn-AMA, or Ni-AMA at 512 µg/mL. However, Co-AMA showed

considerable cytotoxicity and reduced cell viability by 50% at this concentration. These data have been incorporated into the manuscript and are described in the result section (Lines 331-336 and Fig S14)

The structure-activity relationship studies were not exhaustive and were not envisioned to be. However, they were intended to demonstrate that the NikA-ZnAMA cocrystal structure could guide improved NikA binding as a proof-of-principle.

To showcase the application of Zn-AMA *in vivo*, we tested it in a *Galleria mellonella* model. Our data demonstrated that while Zn-AMA has no antimicrobial activity *in vitro*, it can significantly enhance the survival of *G. mellonella* infected with either *K. pneumoniae* or methicillin-resistant *S. aureus*.

We then validated that the effect of Zn-AMA was dependent on urease and showed that a *ureC::tn* mutant had a probability of survival similar to the Zn-AMA treated wild-type bacteria. Furthermore, Zn-AMA had no effect on the survival of *ureC::tn* mutants.

These data validate that urease is a virulence determinant in an animal model and that Zn-AMA can attenuate virulence. The data are presented in the results section (Lines 337-348 and Fig 7.)

Therefore, I'm unable to recommend this manuscript for publication in Nat Com at this stage.

Other comments and suggestions:

AMA as a urease inhibitor:

AMA demonstrated significant inhibition of urease activity as noted on line 134, which would have been an intriguing avenue for further exploration. Given that AMA and Zn-AMA appear to operate on separate pathways, the potential synergistic relationship between these two compounds could have been an interesting area of study. This has already been explored in the case of AHA and Zn-AMA, so it would have been instructive to see a similar examination conducted for AMA and Zn-AMA.

We re-examined the effect of AMA on *K. pneumoniae* in artificial urine and found urease was not fully inhibited. Adding casamino acids to the medium enhanced growth and the AMA-treated cells ultimately showed urease activity, in contrast to the Zn-AMA treated cells (Fig 2 and Fig S2). However, this was not the case in *S. aureus*, which still showed urease inhibition in the presence of both Zn-AMA and AMA (Fig S2). As the focus of this manuscript was *K. pneumoniae*, we did not explore this further and plan to investigate this in a future study.

Mode of action for Zn-AMA:

While Zn-AMA clearly binds to nikA, lending credibility to claims of it being the intended target, the decrease in biological activity when Zn-AMA's binding affinity for nikA is enhanced raises questions. It suggests the possibility that Zn-AMA may have an alternative or additional target, which may be worth exploring in further research.

Our previous work has shown that these analogs have different metal binding habits, such as decreased Zn affinity and selectivity (DOI: 10.1021/acsomega.1c05757). Differences in Zn-affinity could influence the analog's activity in various ways and complicate the determination of their cellular fate. As this was meant to be a proof-of-principle study, we believe that investigating this avenue of each analog is outside the scope of the study.

NikA as a therapeutic target:

The manuscript falls short in substantiating why nikA should be considered a compelling therapeutic target. Questions about the effects of nikA knockout remain unanswered. It's unclear whether nikA mutants can establish an infection or the potential impact this could have *in vivo*. While the importance of Ni-dependent enzymes is mentioned, there's a noticeable lack of detailed information

about *nikA* itself, which is crucial given that it's proposed as your target. Thus, more specific insights about *nikA* should be incorporated.

The downstream effects of NikA ultimately lead to the loss of urease activity. As mentioned above, we assessed the impact of a *ureC::tn* mutant of *S. aureus*, showing that it is important for infection in an animal model. As an initial proof of concept, we believe these data support Ni-homeostasis as an antimicrobial target.

Reviewer #3 (Remarks to the Author):

In this manuscript, Sychantha et al. report that a fungal metallophore (aspergillomarasmine A, AMA) has high affinity for divalent cations, including Zn²⁺, Ni²⁺, and Co²⁺. The use of AMA alone or in combination with the aforementioned metallic cations resulted in inhibition of two well-characterized nickel-containing enzymes, urease and [NiFe] hydrogenases in various organisms. Urease activity was inhibited by AMA in *K. pneumoniae* and *S. aureus* (as shown by measuring enzyme activity), resulting in lower levels of urease-generated struvite crystals in both pathogens. Likewise, (H₂-producing) [NiFe] hydrogenase activity was found to be inhibited in *K. pneumoniae* and *E. coli*. Using a variety of biochemical approaches, the authors do demonstrate that Zn-AMA and Co-AMA bind to NikA, the periplasmic component of the NikABCDE nickel transporter, thereby competing with the natural nickelophore L-His. This is a significant finding. They further characterized the NikA-Ni-AMA binding by X-ray crystallography. Finally, they hypothesized that AMA could be used as a novel antibacterial agent against Ni-requiring pathogens, by inhibiting Ni-enzymes.

The reviewer agrees with the authors that nickel sequestration is a promising avenue to combat antibacterial-resistant pathogens. While this avenue has been already explored by several research groups, the novelty herein is the use of the fungal metallophore AMA. The study is interesting, and contains some significant work, the bulk of the methods used are adequate, and the conclusions are largely supported by results. However, the authors failed to provide any convincing results that would link the use of AMA to growth inhibition and/or virulence attenuation of the various pathogens they study here (e.g., *S. aureus*, *K. pneumoniae*). That addition would make the study complete.

The fact that AMA inhibits Ni-enzymes suggests it might have an effect on the growth of Ni-requiring pathogens. Certainly, growth conditions exist that rely on specific hydrogenases and on urease. Results on *K. aerogenes* reported herein are promising, however they are preliminary, and AMA needs to be tested on other pathogens used in this study (*S. aureus*, *S. Typhimurium*, *K. pneumoniae*), including MDR strains, if possible.

We thank the reviewer for their thorough reading of the manuscript and constructive comments. As indicated in our response to reviewer 2 above, we have validated the efficacy of Zn-AMA as an anti-virulence agent in *G. mellonella* model. Infection models were carried out with *K. pneumoniae* and methicillin-resistant *S. aureus*. We further validated urease as a virulence determinant by showing *S. aureus* USA300 *ureC::tn* mutant was less pathogenic than wild-type, and that the antivirulence activity of Zn-AMA depends on urease (Lines 343 – 346) Fig 7.

A main concern is about the effect of AMA on [NiFe] hydrogenases; I believe this avenue was not properly studied (see below). What type of hydrogenase is inhibited and certainly by how much is not clear. So, my overall impression of this work is mixed, and the study needs to be more thorough.

Major comments

There is lack of basic understanding of hydrogen metabolism, hydrogenases and even nickel homeostasis as a whole throughout the manuscript. Some background needs to be introduced here. Even though the term "hydrogenases" collectively refer to enzymes that can either produce or use H₂, H₂-uptake (oxidizing) hydrogenases and H₂-producing hydrogenases are vastly different enzymes, both in structure and activity, and should be considered as two different types of enzymes, based on their respective contribution, e.g., H₂ -uptake hydrogenases, such as Hya (Hyn) or Hyb, are respiratory enzymes that generate electrons (to feed the ETC) and protons (to generate PMF and ATP),

whereas H₂-producing hydrogenases, such as Hyc and Hyf, are integral parts of larger complexes (FHL and FHL-2), whose main role is to dispose of reductant. The number of hydrogenases within a given organisms, and the type of reactions they perform, is of paramount importance for both growth and virulence in the case of pathogens. Yet, throughout the text, Sychantha et al. fail to specify which type and how many [NiFe] hydrogenases are present in the organisms they study, always referring to "[NiFe] hydrogenase" as if it was one type of enzymes, performing one type of reaction. This lack of precision/accuracy results in misleading statements, incomplete experimental designs, and partial conclusions, as well as a discussion that does not mention anything about hydrogenases!

We acknowledge the reviewer's concerns over the manuscript's oversimplification of the classification and function of [NiFe] hydrogenases.

We have addressed these comments and included relevant *background* on the different reactions catalyzed by [NiFe] hydrogenases and their importance (Lines 80 – 86). Furthermore, to better reflect the hydrogenase complex relevant in *K. pneumoniae*, we have updated Fig 1. It now includes all the components of the FHL complex and the reactions relevant to *K. pneumoniae*.

Furthermore, details about the [NiFe]-hydrogenase for *K. pneumoniae* were included in the results (Lines 171-176). We also indicate that *S. aureus* does not encode [NiFe]-hydrogenases of any kind. These classifications were made based on analyses using the HydDB web tool (<https://services.birc.au.dk/hyddb/> ; DOI: 10.1038/srep34212).

A paragraph on the relevance of [NiFe]-hydrogenases as potential antimicrobial targets has also been included in the discussion. (Line 364-372)

1. abstract, line 35: "critical enzymes involved in colonization and persistence, including urease and [NiFe]-hydrogenase (singular)..."

Hydrogenase was changed to hydrogenases.

2. Introduction, lines 80-81: "Similarly, [NiFe]-hydrogenases catalyze the oxidation of hydrogen (H₂) under anaerobic conditions to generate energy in nutrient-poor environments, as observed for *H. pylori* and *S. Typhimurium*". There are many inaccurate statements in this sentence: first, this suggests that all [NiFe] hydrogenases are H₂-uptake enzymes, which is not the case. Second, anaerobic conditions are not required for H₂-uptake [NiFe] hydrogenases to be active. Third, the two organisms cited as examples could not be more different, as *H. pylori* only possess one (H₂-uptake) hydrogenase, whereas *S. Typhimurium* contains four, including three H₂-uptake types and one H₂-evolving enzyme.

We acknowledge these concerns and revised the manuscript's introduction. (Lines 81-85)

3. Introduction, line 85: "The enzymatic activities of urease and [NiFe]-hydrogenase depend...".

Hydrogenase was changed to the plural form.

4. Results, lines 161-162: "Dose-response assays were performed on bacteria grown in rich media (LB-glucose) using strict anaerobic conditions under which [NiFe]-hydrogenases are produced. Which [NiFe]-hydrogenases are the authors referring to? Under strict anaerobic conditions, and in absence of added H₂, they are likely to monitor only the H₂-evolving hydrogenase, whose contribution to growth and pathogenicity has been shown in previous studies to be negligible.

We understand the reviewer's concern and have provided a clear explanation in the results section (Lines 175-183). The specific hydrogenase is now referred to and experimental conditions are described. Hydrogen evolution and contribution to virulence is described in the introduction (Line 85). Previous work has suggested that inhibiting H₂-evolving

hydrogenases can protect *K. pneumoniae* from oxidative stress. DOI: 10.1007/s12275-010-0149-z

5. Results, lines 166-167: "The distinct effect of these compounds on [NiFe]-hydrogenase activity was verified with zymographic analysis of *E. coli* cell lysates". Which [NiFe]-hydrogenases are the authors referring to? The M&M section specific to the zymograms study mention the use of H₂ and BV. In this condition, they probably observed H₂-oxidizing activity; if so, corresponding enzymes should be described.

We understand the reviewer's concern. This has been clarified in the results (Lines 186-187) based on ref. DOI: 10.1186/1471-2180-12-134

6. Methods, lines 397-406, section on "Whole-cell [NiFe] hydrogenase assay". References are missing, and probably for a good reason; I have not seen this protocol before, and the aim of this assay is not clear. The experimental design chosen by the authors (anaerobic growth with no H₂ provided) suggests that they expect to monitor H₂-producing hydrogenases (Hyc or Hyf), however these usually have negligible activity, and they have been shown to be the least important both in terms of growth and virulence of pathogenic bacteria (see previous published work on *S. Typhimurium* hydrogenases). In addition, if the goal was to monitor H₂ production, then reduced BV should be used (as electron donor), not oxidized BV---an electron acceptor is not needed for H₂ production assays. In summary, it is not clear to me what they are trying to achieve here, but these results are incomplete at best.

The assay is a modified method previously described by the Zamble lab for *E. coli* (10.1074/jbc.RA119.008101). It was intended to promote the fermentation of glucose for mixed acid production under anaerobic conditions. Under this condition, the reduction of benzyl violagen by an H₂-producing hydrogenase (Hyc/Hyd-3) could be detected, which is in line with previous work (Described on Lines 175 -183 and ref DOI: 10.1186/1471-2180-12-134, DOI: 10.1128/jb.164.3.1324-1331.1985, and DOI: 10.1007/s00203-011-0726-5). The observation that BV could be reduced by this enzyme provided a convenient readout for the NikA-dependent activity of Hyc.

The references have been included in the manuscript.

7. In the discussion section, the authors do not discuss the potential and limitations of using AMA alone or with metals to inhibit hydrogenase activity/activities. In fact, hydrogenases are not mentioned at all in the discussion section.

A paragraph on the relevance of [NiFe]-hydrogenases as potential antimicrobial targets has also been included in the discussion. (Line 364-372)

Minor comments:

A number of sentences are vague; this includes the title of the manuscript, paragraph headings, statements, figure legends etc; sometimes there is no transition between paragraphs or conclusions. Also, the authors describe nickel homeostasis in various organisms (*K. pneumoniae*, *E. coli*, *S. aureus*), however often it is not clear which one they refer to when reporting results or describing previously described mechanisms. A few examples:

1. The title "Targeting bacterial nickel transport inhibits virulence-associated Ni-dependent enzymes" reads like a review title, and does not let readers expect anything novel, since (i) inhibiting nickel transport is expected and has already been shown to result in Ni-enzyme inhibition; and (ii) several Ni-enzymes (hydrogenases and urease) have already been shown to be required for growth and virulence in several pathogens. The novelty here is the use of the fungal metallophore aspergillomarasmine A to inhibit Ni-enzymes and Ni-requiring pathogens, so this should be included in the title.

An alternative title has been chosen: Targeting bacterial nickel transport with aspergillomarasmine A suppresses virulence-associated Ni-dependent enzymes.

2. Abstract, lines 38-39 " AMA may block Ni²⁺ uptake and attenuate Ni-dependent enzymes as a potential anti-virulence strategy" does not read well

An alternative statement was included: The fungal metallophore aspergillomarasmine A (AMA) shows a narrow specificity for Zn²⁺, Ni²⁺, and Co²⁺. This specificity allows AMA to block the uptake of Ni²⁺ and attenuate Ni-dependent enzymes, offering a potential strategy for reducing virulence. (Lines 37-40)

3. Abstract, lines 51-52: " These findings...in the chemotherapeutic treatment of bacterial infections that depend upon Ni²⁺ as a micronutrient". Bacteria depend upon Ni as a micronutrient, bacterial infections do not.

This statement has been corrected.

4. Introduction, lines 86-91 "The ABC-type transporter (NikABCDE) imports the metal from the extracellular environment..." which organisms are the authors referring to in this statement? It comes right after describing the role of hydrogenase in *H. pylori* and is written as if NikABCD was the universal way of importing Ni²⁺ into the cytoplasm; however, some bacteria, including *H. pylori*, use other types of high affinity transporters (NixA, Niu). Please be more specific.

Specificity was incorporated (Lines 88-90).

5. Introduction, line 59 . "linking the role of Ni-dependent enzymes to the colonization..., not in the colonization..."

This was corrected.

6. Introduction, line 61" *Salmonella enterica* serovar Typhimurium, or *Salmonella enterica* Typhimurium" when introduced for the first time.

This was corrected.

7. Introduction, lines 64-65. Please provide one or several references for the statement on calprotectin.

A reference was added.

8. Fig. 1. "The role of nickel in virulence". This Fig. title is vague. Which organism are they referring to? Gram negative obviously, since an outer membrane is depicted? *E. coli*? *K. pneumoniae*? Please be more specific.

This was corrected.

9. Fig. 3. "Bacterial nickel uptake is inhibited by competitive binding to the periplasmic Ni-binding protein NikA". Which organism are the authors referring to?

This was corrected.

10. Results, lines 120-121: "The K_d value for Ni²⁺ was estimated to be 80 pM, 2-fold higher than AMA's affinity for Zn²⁺ and is consistent with the Irving-Williams order of ligand strength for these metals (Fig. 2A and Fig. S1)."First, the authors report a K_d value of 80 pM for Ni-AMA and 200 pM for

Zn-AMA, which is 2.5-fold higher, not 2-fold higher. Of concern, both K_d values are in the pM range (text, and Fig. 2A) whereas they are reported in the nM range (ITC results, Fig. S1).

Our apologies for the confusion. The values shown in the ITC results were typos and meant to be reported as pM values.

11. Results, lines 143-144: "These results confirm our hypothesis and the ability of AMA to suppress urease activity while also suggesting a mechanism beyond Ni^{2+} sequestration". I do not understand this sentence. Please rephrase.

This has been corrected, stating: "Overall, these data indicate that Zn-AMA suppresses urease activity without interacting with Ni^{2+} ions, inferring that it may interact with a specific protein target." (Lines 154-156).

12. Results, lines 148-157. The authors discuss two types of mechanisms by which Zn-AMA and AHA could inhibit urease activity. One involves "synergism", whereas the other would be "additive effect". In my opinion, synergism and additive effect mean the same.

These terms are distinct. Additive effects for two compounds equal the sum of their individual effects, whereas synergistic effects exceed the sum of their individual effects.

13. Results, lines 150-151 "Zn-AMA inhibited urease indirectly by interfering with urease maturation in *K. pneumoniae* by disrupting Ni^{2+} uptake" does not read well.

This has been corrected

14. Discussion, line 317: what is an "untargeted" mechanism?

The statement was intended to communicate that it does not engage with a specific protein target. This statement was removed for clarity.

15. Methods, lines 387-395. Urease activity in cell-free extracts has to be standardized to protein concentration. Even when using the same method to break cells, one cannot expect the lysis to yield the same amount of protein, including urease; this is likely to have an effect on the final enzyme activity levels.

We agree with the reviewer. This experiment has been repeated and normalized to protein concentration. A revised figure is included (Fig S6) and the method updated (472-473).

16. Methods, line 440: "*K. pneumoniae* Nika...was cloned into pET28b..." A gene (nikA) (not a protein) is cloned into a plasmid.

This has been corrected

17. Methods, line 444 " β -galactoside" needs to be corrected

This has been corrected.

18. Methods, line 455, same as above "Nika cloned into pET-28b..." A gene (nikA) is cloned, not a protein. Also, I believe a "was" is missing

This has been corrected.

19. Fig. 3. "Bacterial nickel uptake is inhibited by competitive binding to the periplasmic Ni-binding protein Nika". Which organism are the authors referring to?

The organism's name has been added.

20. Results presented in Fig S3 and Fig S5 are confusing, Fig. legends are vague and do not give an accurate description of the experiment. Fig S3 legend: "Dose-response analysis of ZnCl₂ on urease activity of *K. pneumoniae*" does not mean anything, please rephrase. Fig. S3 shows the effect of ZnCl₂ on urease activity, but Fig S3 legend states "Urease activity was determined in artificial urine supplemented with 2-fold dilutions of AMA and its metal complexes at 37 °C for 24 hours." Did the authors study the effect of ZnCl₂ only on urease activity in *K. pneumoniae*? The effect of Zn-AMA? The effect of Zn in presence of AMA and other metal complexes?

We apologize for the confusion, Fig S3. Shows the effect of ZnCl₂ alone. The legends have been clarified.

21. Fig. 1 Legend: "Ni-dependent enzyme" should be plural ("enzymes").

This has been corrected.

22. Fig. 9 should be Fig. S9.

23. Fig. S9 Legend. "Relative urease activity was measured as a change in resulting from a pH change in the media" does not read well and needs to be rephrased. I believe "absorbance at 550 nm" is missing". Also, "It was detected with an increase in absorbance at 550 nm using the pH indicator phenol red" is confusing. Is there still an increase in absorbance when authors report 0% relative urease activity? The difference in absorbance at 550 nm between T0 and T24 is what matters.

This has been corrected.

24. Fig S1 legend refers to (A) and (B), but there is no (A) and (B) shown on Fig S1.

A and B have been added.

25. Fig S3 legend "Dose-response analysis of ZnCl₂ on urease activity of *K. pneumoniae*" does not mean anything, please rephrase.

This has been corrected.

26. Fig. S5 Legend. FIG S5. "Dose-response analysis of AMA and Zn-AMA in cell-free extracts of *K. pneumoniae*" does not mean anything, please rephrase. And "*K. pneumoniae*" needs to be corrected.

This has been corrected.

27. A gel showing purified KpNikA and EcNikA recombinant proteins should be shown in the supplement.

SDS-PAGE gels of both proteins have been included in the supplemental figures.(Fig. S15)

We appreciate the thorough evaluation made by the reviewer We have addressed their comments and made the recommended corrections to the manuscript.

Reviewer #1 (Remarks to the Author):

The authors have addressed nearly all the concerns expressed in the original review. The demonstration that Zn-AMA or Co-AMA can bind to NikA and inhibit nickel uptake is a significant result. A few remaining or additional issues are indicated below:

Lines 91 and 92: rewrite "Ni uptake by NikA is promiscuous" (perhaps replace the last word with widespread unless the authors mean NikA is promiscuous in terms of what it transports) and "L-histidine is a significant source of Ni" (an organic ligand cannot be the source of a metal ion).

Line 94: Fig 1 is said to show Ni trafficking in *K. pneumoniae*; however, urease activation in this microorganism uses UreDEFG (not UreEFGH) and SlyD has not been shown to facilitate this process in this species.

Line 119: fix the spelling for geometry and replace the mention of Fig 1A with Fig 2A.

Line 124: The text incorrectly states that the K_d for Ni was 2-fold greater than that for Zn.

Line 126: Fig S1B was used to determine the 80 pM K_d for Ni, not both that value and the 200 pM K_d of Zn as shown. The value for Zn was obtained previously and should be cited in the legend. Curiously, the 200 pM value appears to be taken from the JBC paper published in 2021, but a more recent ACS Omega paper from 2022 by these authors shows a value of 100 pM. Why was the older value utilized?

Line 134: More accurately, the authors "used whole-cell urease-dependent changes in pH as a readout to assess Ni availability in the cytoplasm".

Line 154: Do the authors have an explanation for why low concentrations of AMA lead to an increased number of crystals in Fig. 2C?

Line 154: Fig S5 should specify the meaning of the concentrations shown above the micrographs.

Line 158: Following the word "complexes", the word "its" is unclear and has number agreement issues. The sentence could be mistakenly interpreted as AMA and AHA having a direct interaction and should be rewritten.

Line 170: Fig 2F should be Fig 2E.

Line 182: Fix the spelling of viologen.

Line 257: In Fig 5B, it is still unclear why the mesh is shown around His416 when the legend says it is for the Ni-AMA complex. This issue was mentioned in the earlier review and the response to this point was incorrect. In the legend to Fig 5C, the line "Atoms with rotatable bonds are colored blue" is confusing (blue indicates nitrogen atoms) and is not necessary; delete.

Line 263: I question the use of the phrase "non-polar cage" given the extensive electrostatic interactions mentioned in the next sentence and the solvent molecules.

Line 304: What is meant by "lack of activity of Ni complexes with AMA"? Should this sentence be changed to "because Ni-AMA does not affect the pH increase due to cellular urease"?

Line 314: AMA should be Zn-AMA.

Line 434: clarify the meaning of Strep-II-tag and TEV by citing references or providing the manufacturer information.

Line 443: change to using a subscript for OD600.

Line 452: clarify pen/strep.

Line 465: The authors determined pH changes, not urease activities for whole cells.

Line 603: table S1, not table 1.

Reviewer #2 (Remarks to the Author):

The revised manuscript by Sychantha et al. represents a significant improvement over its predecessor. The authors have effectively addressed the primary concerns highlighted in the initial review. In the current version, the manuscript convincingly establishes the validity of nika as a therapeutic target and underscores the efficacy of Zn-AMA as a potent inhibitor of this target. However, the SAR data presented in this study appears to suggest a more complex MOA for Zn-AMA. The presented work is both innovative and offers an alternative approach to combatting resistant bacterial infections, specifically *K. pneumoniae* and *S. aureus*, which are on the WHO critical list.

Comments and concerns:

Line 112 and (408-410): This sentence/statement is slightly misleading. The SAR work presented here is minimal and raises more questions than answers. It does highlight some of the important areas of the molecule, but it doesn't provide a basis for the design and synthesis of more potent analogs. As there was no correlation between Kd and efficacy, it would be difficult to design more potent analogs from this data.

The authors note (lines 325-327) "That while Nika-AMA interactions can be improved, other factors significantly influence biological activity". I agree with this statement, and before these other factors are determined, the statements on lines 112 and 408-410 are a little strong.

Line 119: Figure 1A should be Figure 2A.

Lines 143-144: a reference to a figure would be helpful for this sentence.

Lines 151-154: Zn-AMA does appear to prevent/inhibit struvite production. Considering struvite production facilitates biofilm formation, it would be informative to investigate if Zn-AMA can inhibit biofilm formation. As biofilms are highly problematic in urinary catheter-related infections, this could be a valuable additional use for compound Zn-AMA.

Line 154: Figure S5 does not specify what the treatments for the top and bottom rows are.

Line 302: Were these synthetic AMA analogs prepared in-house? If so, please include methods and characterisation.

Reviewer #3 (Remarks to the Author):

[No comments for authors]

Reviewer #1 (Remarks to the Author): **Revisions highlighted in yellow in the text**

The authors have addressed nearly all the concerns expressed in the original review. The demonstration that Zn-AMA or Co-AMA can bind to NikA and inhibit nickel uptake is a significant result. A few remaining or additional issues are indicated below:

Lines 91 and 92: rewrite “Ni uptake by NikA is promiscuous” (perhaps replace the last word with widespread unless the authors mean NikA is promiscuous in terms of what it transports) and “L-histidine is a significant source of Ni” (an organic ligand cannot be the source of a metal ion).

We thank the reviewer for raising this point. This statement was intended to convey that the Nik system uses exogenous histidine that could be derived from different sources and that a dedicated bacterial nickelophore associated with the Nik system is unknown. **Lines 91 and 92** have been changed according to the reviewer's recommendation.

Line 94: Fig 1 is said to show Ni trafficking in *K. pneumoniae*; however, urease activation in this microorganism uses UreDEFG (not UreEFGH) and SlyD has not been shown to facilitate this process in this species.

We have confirmed this and made the recommended change.

Line 119: fix the spelling for geometry and replace the mention of Fig 1A with Fig 2A.

Line 119 has been corrected.

Line 124: The text incorrectly states that the K_d for Ni was 2-fold greater than that for Zn.

We have corrected the statement and have now indicated that the K_d for Ni is 2-fold lower than Zn on **Line 123**.

Line 126: Fig S1B was used to determine the 80 pM K_d for Ni, not both that value and the 200 pM K_d of Zn as shown. The value for Zn was obtained previously and should be cited in the legend. Curiously, the 200 pM value appears to be taken from the JBC paper published in 2021, but a more recent ACS Omega paper from 2022 by these authors shows a value of 100 pM. Why was the older value utilized?

No significant difference was found in the K_d of Ni when either Zn K_d values were used to calculate it. Furthermore, we have found that AMA's K_d for Zn drifts typically between 100 – 200 pM. It has not been possible to obtain a more precise value using the PAR-based competition assay.

Line 134: More accurately, the authors “used whole-cell urease-dependent changes in pH as a readout to assess Ni availability in the cytoplasm”.

We thank the reviewer for the suggestion. **Line 134** has been changed accordingly.

Line 154: Do the authors have an explanation for why low concentrations of AMA lead to an increased number of crystals in Fig. 2C?

It is unclear why more crystals appear present at low concentrations of AMA. As this trend does not track with the pH of the medium, there appear to be other unknown factors influencing crystal nucleation under these experimental conditions. This is acknowledged in lines 154 -156.

Line 154: Fig S5 should specify the meaning of the concentrations shown above the micrographs.

We thank the reviewer for identifying this. Figure S5 has been changed to denote which micrographs correspond to AMA and Zn AMA.

Line 158: Following the word “complexes”, the word “its” is unclear and has number agreement issues. The sentence could be mistakenly interpreted as AMA and AHA having a direct interaction and should be rewritten.

Line 158 (Now 166) has been improved and clarifies that a chemical-chemical interaction between Zn-AMA and AHA was investigated.

Line 170: Fig 2F should be Fig 2E.

The typo has been corrected.

Line 182: Fix the spelling of viologen.

The typo has been corrected.

Line 257: In Fig 5B, it is still unclear why the mesh is shown around His416 when the legend says it is for the Ni-AMA complex. This issue was mentioned in the earlier review and the response to this point was incorrect. In the legend to Fig 5C, the line “Atoms with rotatable bonds are colored blue” is confusing (blue indicates nitrogen atoms) and is not necessary; delete.

The 2mFo-DFc and mFo-Fc maps for His416 provide evidence for this residue's involvement in coordinating the Ni-AMA complex. To clarify the figure, the legend had been updated to accurately describe these maps and the components they are contoured around.

The statement about blue bonds has been removed.

Line 263: I question the use of the phrase “non-polar cage” given the extensive electrostatic interactions mentioned in the next sentence and the solvent molecules.

To avoid future confusion, we have removed “non-polar” cage from **line 263 (now 270)**.

Line 304: What is meant by “lack of activity of Ni complexes with AMA”? Should this sentence be changed to “because Ni-AMA does not affect the pH increase due to cellular urease”?

We agree with the reviewer's comment and have changed **lines 308-309 (now line 311)** to their suggestion.

Line 314: AMA should be Zn-AMA.

Line 314 has been corrected.

Line 434: clarify the meaning of Strep-II-tag and TEV by citing references or providing the manufacturer information.

The tag and cleavage sequences were defined, and a reference to the Strep system was included.

Line 443: change to using a subscript for OD600.

The subscript has been inserted.

Line 452: clarify pen/strep.

This has been clarified on **line 460**: 100 µg/mL streptomycin and 100 µg/mL penicillin.

Line 465: The authors determined pH changes, not urease activities for whole cells.

The heading has been corrected.

Line 603: table S1, not table 1.

This has been corrected.

Reviewer #2 (Remarks to the Author): **Revisions highlighted in green in the text**

The revised manuscript by Sychantha et al. represents a significant improvement over its predecessor. The authors have effectively addressed the primary concerns highlighted in the initial review. In the current version, the manuscript convincingly establishes the validity of *nikA* as a therapeutic target and underscores the efficacy of Zn-AMA as a potent inhibitor of this target. However, the SAR data presented in this study appears to suggest a more complex MOA for Zn-AMA. The presented work is both innovative and offers an alternative approach to combatting resistant bacterial infections, specifically *K. pneumoniae* and *S. aureus*, which are on the WHO critical list.

Comments and concerns:

Line 112 and (408-410): This sentence/statement is slightly misleading. The SAR work presented here is minimal and raises more questions than answers. It does highlight some of the important areas of the molecule, but it doesn't provide a basis for the design and synthesis of more potent analogs. As there was no correlation between K_d and efficacy, it would be difficult to design more potent analogs from this data.

The authors note (lines 325-327) "That while *NikA*-AMA interactions can be improved, other factors significantly influence biological activity". I agree with this statement, and

before these other factors are determined, the statements on lines 112 and 408-410 are a little strong.

We thank the reviewer for the feedback. We have changed **Line 112**, which now states: “we explored the structure-activity relationships of Zn-AMA, which validated components of the molecule that are important for NikA interactions.”

We have also softened the language on **Lines 408-410 (Now 414-417)**, stating: “Zn-AMA represents a new example of a metal-based compound with therapeutic potential, setting a precedent for identifying additional analogs with improved biological activity and broad-spectrum activity.”

Line 119: Figure 1A should be Figure 2A.

We thank the reviewer for identifying this typo. The correct figure has been inserted.

Lines 143-144: a reference to a figure would be helpful for this sentence.

Figure S2 has been included at the end of the sentence.

Lines 151-154: Zn-AMA does appear to prevent/inhibit struvite production. Considering struvite production facilitates biofilm formation, it would be informative to investigate if Zn-AMA can inhibit biofilm formation. As biofilms are highly problematic in urinary catheter-related infections, this could be a valuable additional use for compound Zn-AMA.

We appreciate the reviewer’s helpful suggestion. Biofilm assays were conducted at different concentrations of Zn-AMA. The complex suppressed biofilm production in *K. pneumoniae* grown in artificial urine, but not M9 minimal medium. This contrasted with *S. aureus*, whose biofilm production was unaffected in either growth condition, suggesting differences in the biology of its biofilms compared to *K. pneumoniae*.

The correlation of *K. pneumoniae* biofilm production with urease activity and struvite production further supports that Ni²⁺ is important for this organism's virulence and adds further value to Zn-AMA. The data has been included in **Fig 2D and Fig S6**. The results are described on **Lines 157 -165**.

Line 154: Figure S5 does not specify what the treatments for the top and bottom rows are.

Thank you, this has been corrected.

Line 302: Were these synthetic AMA analogs prepared in-house? If so, please include methods and characterisation.

The synthetic analogs were prepared in-house, for which the methods and characterization have been published elsewhere previously (<https://pubs.acs.org/doi/10.1021/acsomega.1c05757>.) The reference has been included on **Line 310**.

Reviewer #2 (Remarks to the Author):

This is the third version of the manuscript by Sychantha et al. I thoroughly enjoyed reviewing this manuscript and commend the authors for diligently addressing all the reviewer's comments and suggestions. I wish the authors the best as they continue with the publication process and eagerly anticipate witnessing the development of this novel strategy. I have no further comments or suggestions for the authors. Good luck!

Minor edits:

Line 39: change Ni^+ to Ni^{2+}

Line 215: change "addition of access" to "addition of excess"

Response to reviewers:

Reviewer #2 (Remarks to the Author):

This is the third version of the manuscript by Sychantha et al. I thoroughly enjoyed reviewing this manuscript and commend the authors for diligently addressing all the reviewer's comments and suggestions. I wish the authors the best as they continue with the publication process and eagerly anticipate witnessing the development of this novel strategy. I have no further comments or suggestions for the authors. Good luck!

Thank you very much for the thorough evaluation of the manuscript. The comments not only helped to improve the story but also provided additional value to the AMA as an anti-virulence agent.

Minor edits:

Line 39: change Ni^+ to Ni^{2+}

Line 215: change "addition of access" to "addition of excess"

Edits have been made to correct these typos.